# The evolution of deep convective systems and their associated cirrus outflows

George Horner[1] and Edward Gryspeerdt[1,2]

[1]Space and Atmospheric Physics Group, Imperial College London, London, UK
[2]Grantham Institute – Climate Change and the Environment, Imperial College London, London, UK

**Correspondence:** George Horner (g.horner20@imperial.ac.uk)

**Abstract.** Tropical deep convective clouds, particularly their large cirrus outflows, play an important role in modulating the energy balance of the Earth's atmosphere. Whilst the cores of these deep convective clouds have a significant short-wave (SW) cooling effect, they dissipate quickly. Conversely, the thin cirrus that flow from these cores can persist for days after the core has dissipated, reaching hundreds of kilometres in extent. These thin cirrus have a potential for large warming in the tropics. Understanding the evolution of air parcels from deep convection, clouds along these trajectories, and how they change in response to anthropogenic emissions is therefore important to understand past and future climate change.

This work uses a novel approach to investigate the evolution of tropical convective clouds by introducing the concept of "time since convection" (TSC). This is used to build a composite picture of the lifecycle of air parcels from deep convection. Cloud properties are a strong function of TSC, showing decreases in the optical thickness, cloud-top height, and cloud fraction over time, thereby driving the latitudinal structure of cloudiness. After an initial dissipation of the convective core, changes in thin cirrus cloud amount were seen beyond 200 h from convection. Changes in cloud are shown to be a strong function of TSC and not simply reflective of latitudinal changes as air moves from the tropics to the extratropics.

Finally, in the initial stages of convection there was a large net negative cloud radiative effect (CRE). However, once the convective core had dissipated, the sign of the CRE flipped and there was a sustained net warming CRE beyond 120 h from the convective event. Changes are present in the cloud properties long after the main convective activities have dissipated, signalling the need to continue further analysis at longer timescales than previously studied.

## 1 Introduction

Deep convective clouds (DCCs) occur primarily in the tropical and subtropical regions. They are characterised by one or more convective cores (cumulus towers), strong updraughts that merge the cores at higher altitudes, and large anvils flowing out from the region of convection (Fan et al., 2013). The radiative forcing from deep convective cores in the tropics can vary greatly depending on the optical thickness and ice water path of the cloud (Berry and Mace, 2014; Hartmann and Berry, 2017). However, the overall total cloud radiative effect (CRE) in the tropics is small (Wielicki et al., 1996). This is due to the cancellation of the large long-wave (LW)

warming that occurs in cloudy regions in the tropics and the significant short-wave (SW) CRE that comes from the high albedo of these same regions. This leads to the CRE in cloudy regions of the tropics being very similar to the CRE in nearby non-convective regions (Hartmann and Berry, 2017; Harrison et al., 1990; Ramanathan et al., 1989).

Whether this cancellation is due to feedbacks or is simply a coincidence is unclear (Kiehl, 1994; Hartmann et al., 2001b). If it is a coincidence, then the behaviour of these deep convective regimes may change in a warming world or under different aerosol concentrations. This cancellation is the sum of large SW and LW components. Therefore any small change

in either of these components could potentially have a large effect on the overall CRE. Understanding the lifecycle and drivers of deep convection is thus important to understand the sensitivity of these regimes.

Whilst the convective cores alone have a negative CRE, their lifetimes are relatively short in comparison to the outflowing anvil cirrus. Hartmann et al. (2001b) found that the top-of-atmosphere (TOA) radiative forcing of DCCs ranged between $+20$ and $-119\,\mathrm{W\,m^{-2}}$, with the optical depth of tropical DCCs ranging from 1 to 60 in the upper troposphere. On the other hand, the detrained thin cirrus from the anvil of the top of convective core can persist for up to 5 d (Luo and Rossow, 2004), cover large spatial extents, and have a significant warming contribution to the CRE (Protopapadaki et al., 2017; Wall et al., 2018).

The sign of the radiative forcing from cirrus in the tropics is highly dependent on the optical thickness of the cloud ($\tau_c$). Choi and Ho (2006) found that thin cirrus between 10 and 12 km with an optical depth of $\tau_c < 10$ have a warming effect, whilst cirrus with a $\tau_c > 10$ have a cooling effect. Koren et al. (2010) showed that the cloud-top height also plays a vital role in the sign of the forcing from cirrus. They showed that if the cirrus cloud-top height is increased and the optical depth decreased, there is potential for much greater warming.

The lifecycle of deep convective cores (DCCs) and their associated anvil and thin cirrus outflows has been investigated in numerous prior works, in both observations and models. Using observational data, Luo and Rossow (2004) found that 50 % of tropical cirrus clouds originated from convection. They found that the detrained cirrus had, on average, longer lifetimes than the in situ cirrus, at around 30 h compared to 19 h for in situ. This suggests some mechanism for sustaining water vapour at the detrained cirrus height that allows for longer lifetimes. Using ground-based data and satellites, Mace et al. (2006) found that 47 % of cirrus observed over Manus Island in the western Pacific originated from deep convection occurring within the past 12 h. Many other modelling and observational studies also show that at least 50 % of tropical cirrus originate from convection (Massie et al., 2002; Gehlot and Quaas, 2012; Riihimaki et al., 2012). Salathé and Hartmann (1997) showed that upper tropospheric humidity is sustained for up to 5 d from convection. This is consistent with the cirrus decay found by Luo and Rossow (2004).

Garrett et al. (2005) looked at the detailed evolution of a single thunderstorm anvil cirrus cloud in Florida. They found that the advected outflow was separated into two cloud layers: (1) a cirrus anvil at $-45\,^{\circ}\mathrm{C}$ lying below (2) a thin tropopause cirrus (TCC) at $-70\,^{\circ}\mathrm{C}$. The TCC lifetime was sustained longer than that of the anvil cirrus, potentially due to the anvil cirrus shielding the TCC from terrestrial radiation, as proposed by Hartmann et al. (2001a). Schwartz and Mace (2010) used CloudSat and CALIPSO data to examine this mechanism. For a complete overview of the different cirrus types and their origin, Krämer et al. (2016) synthe-

sised multiple field campaigns, simulations, and model results looking at cirrus microphysics by splitting up in situ, detrained, liquid origin, and tropical tropopause layer (TTL) cirrus.

Studies investigating the model evolution of cirrus found broadly similar results to those found in observations. Gehlot and Quaas (2012) saw a rapid decay (6–12 h) of deep convection from the peak and an associated growth of cirrus in the ECHAM5 model. After 1 d they found a decay of the cirrostratus, likely associated with a loss of cloud water due to larger particles that have a large sedimentation mass flux. Jensen et al. (2018) investigated the detrainment of ice crystals from the tops of deep convection in a cloud-resolving model (CRM). They found that most large ice crystals (> 200 µm in diameter) fall out of the anvils within 2 h of formation. The cirrus decay was much slower, occurring over 120 h, from a maximum cirrus cloud fraction of 0.4 to 0.15 after 120 h. They also looked at changes under increased sea surface temperatures (SSTs), finding anvil cirrus clouds to have increased cloud fractions and higher cloud tops, leading to a positive cloud feedback.

When considering the CRE of the evolution of the convective system, it is the balance between the LW and SW over the entire lifetime of the cirrus in the tropics that, if perturbed, could lead to large changes in the overall net CRE. Most tropical radiative studies are confined to models. Looking at the CRE along Lagrangian trajectories in simulation, Gasparini et al. (2021) found that large-SW and large-LW values dominate close to convection, as expected. The net CRE oscillates close to zero and is a function of the solar insolation. In their simulations most convection occurs in the early morning, and there is a peak in net negative CRE at peak solar insolation when SW cooling dominates. There is a slight warming at 15 h when the anvil has thinned sufficiently and the solar insolation decreased enough for the LW CRE to dominate.

The factors that control the radiative evolution of the decaying cirrus have also been investigated in model studies. Gasparini et al. (2019) found that at high (> $100\,\mathrm{g\,m^{-2}}$) IWP (ice water path) values, or above the 80th percentile, the net CRE is negative. Beyond this, at low IWP and correspondingly low cloud fractions above 12 km, the net CRE switches signs to a small positive value. This switch occurs when the $\tau_c$ is around 4, roughly where precipitation stops occurring. This is a lower value for $\tau_c$ than found in observations by Choi and Ho (2006). As the CF and IWP decrease, the CRE tends to zero. The switch from warming to cooling happens at around 6 h from convection in Gasparini et al. (2019). They generally found that small-scale microphysical and dynamical processes are responsible for determining the lifecycle of the detrained anvil cloud. Current global circulation models (GCMs) have horizontal and vertical resolutions that are too coarse to accurately reflect the dynamical processes controlling anvil–cirrus evolution (Wall and Hartmann, 2018).

Most previous work focused on individual mesoscale studies and capped the analysis at shorter timescales in the region of 120 h. In contrast, this paper presents a method to run a Lagrangian trajectory analysis across the entirety of the tropics at unbounded trajectory lengths at a low computational cost, leading to changes being found in the cloud properties at timescales far beyond 120 h after the initial detrained cirrus from convection has dissipated and in situ cirrus has formed. Used in conjunction with lidar and radar data, the vertical evolution of the cloud properties is characterised as a function of time since convection (TSC). The approach in this work builds a composite picture of the lifecycle from tropical convective into thin cirrus. This paper also investigates how TSC is a function of latitude and shows that the changes in the cloud properties along trajectories are a strong function of TSC and are not simply reflective of latitudinal changes as air moves from the tropics to the extratropics. This paper also considers the radiative evolution for just the high clouds along the trajectories.

## 2 Method

### 2.1 Data

Observational data from the 3-hourly International Satellite Cloud Climatology Project H (ISCCP-H) dataset at $1 \times 1°$ are used to define locations of deep convection (Rossow et al., 2017). ECMWF ERA5 reanalysis wind fields are used in the trajectory analysis (Hersbach et al., 2018) and to characterise cloud properties. To examine the evolution of the radiative properties, the CERES SYN1deg L3 LW and SW TOA fluxes are used (NASA/LARC/SD/ASDC, 2017). The CERES SYN1deg product combines MODIS and geostationary satellite data to provide global coverage at a $1 \times 1°$ resolution and 1-hourly temporal resolution.

To illustrate the vertical cloud evolution and microphysical properties, the DARDAR dataset, an ice cloud retrieval product that combines measurements from the CloudSat radar and CALIPSO lidar (Delanoë and Hogan, 2008a; Sourdeval et al., 2018), is used. The ice crystal number concentration, $N_i$, is highlighted from the DARDAR data due to the complementary information provided about cloud history compared to the large-scale properties from ISCCP (Krämer et al., 2016). The period of study in this paper is 2008–2010 inclusive.

### 2.2 Convective-core identification

In order to track the time since a parcel of air last experienced convection, convective cores were defined and identified in the satellite data. This work uses the global weather states in Tselioudis et al. (2021), which uses the ISCCP-H $1 \times 1°$ daytime and nighttime 3-hourly dataset (Rossow et al., 2017). The ISCCP-H differs from prior ISCCP products in numerous ways, most importantly by improving the

**Table 1.** ISCCP-H cloud-regime centroid values for the CTP, $\tau_c$, and CF from Tselioudis et al. (2021). The DCC regime is highlighted in bold.

| Regime | CTP/hPa | $\tau_c$ | CF/% |
|---|---|---|---|
| **Deep convective cores** | **242.6** | **10.5** | **99.5** |
| Mid-latitude storm | 433.6 | 10.4 | 99.2 |
| Thin high cirrus | 316.3 | 1.2 | 79.9 |
| Polar | 395.6 | 2.2 | 84.5 |
| Middle–top clouds | 606.9 | 9.5 | 97.2 |
| Fair weather | 645.1 | 3.2 | 40.0 |
| Shallow cumulus | 840.1 | 4.0 | 79.6 |
| Stratocumulus | 725.5 | 6.3 | 90.7 |

spatial resolution from 2.5 to 1°. A full list of the differences in the data products is given in Rossow et al. (2017). The ISCCP-H dataset is separated into seven distinct cloud regimes by calculating the nearest neighbours of each grid box and clustering them into the separate regimes depending on their cloud fraction, albedo, and cloud-top pressure. These cloud regimes, and their associated centroid values, are shown in Table 1. The albedo is calculated from the optical thickness using Eq. (1),

$$\alpha = \frac{\tau_c^{0.895}}{\tau_c^{0.895} + 6.82},$$ (1)

as defined in the ISCCP simulator (Klein and Jakob, 1999). For a full description of the ISCCP-H weather states, see Tselioudis et al. (2021).

This study is focused on the tropical region of 30° S–30° N and only considers the deep convection defined within the tropics, whereas the weather states in Table 1 are defined globally. Note that the value of 10.5 for the optical-depth centroid for the deep convection is typically lower than the optical depth of deep convective cores in high-resolution satellite data. The ISCCP data used here are at $1 \times 1°$, and therefore the bins will have a lower average optical depth across the whole grid box.

Additional conditions are applied to isolate the convective cores: a $\tau_c > 8.5$ (albedo > 0.5) and a cloud-top temperature (CTT) < 220 K. Only the very brightest, thickest cores of the convective clouds are categorised as DCCs. If these conditions are not imposed, then thick anvil cirrus are included as part of the convective cores, and the ability to investigate their temporal development is reduced.

The mean relative frequency of occurrence (RFO) for the convective cores for the years 2007–2010 is shown in Fig. 1. Most of the convection occurs in the Maritime Continent, over central Africa and South America, and along the intertropical convergence zone (ITCZ). This work is only concerned with tropical convection as defined in the region between the red boundaries (30° S–30° N).

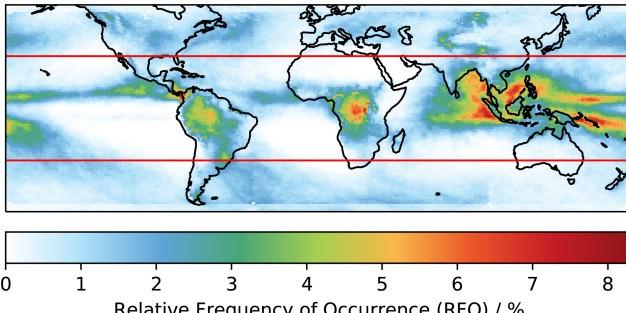

**Figure 1.** Relative frequency of occurrence (RFO) of the convective-core regime from between 2007 and 2010. Red bands show the boundaries of the tropics (30° S–30° N).

## 2.3  Time since convection

The ECMWF ERA5 reanalysis wind speeds at 0.25° are used in this study (Hersbach et al., 2018). A pressure-averaged wind field of between 200 and 300 hPa is used to advect the convective cores forward in order to best reflect the range of heights that the convective cores, and the subsequent detrained and in situ cirrus, exist in. This is the same logic as applied in Luo and Rossow (2004), where vertical movement of air is not considered, but they use a pressure-averaged wind field between 200 and 500 hPa. The decision to use a higher-altitude average wind field is addressed in Sect. 2.5.

The advection process is as follows. All the pixels in the tropics are advected forward 1 h in time according to the wind field. The TSC now has the value of $T + 1$ h since convection. In locations where DCCs are present, the TSC is reset to zero. This advection process repeats, each time resetting locations of DCCs to zero TSC. Figure 2 shows a schematic diagram of the process in one dimension.

The TSC algorithm runs at $0.1 \times 0.1°$ resolution. Running the TSC code at $0.1 \times 0.1°$ is necessary to ensure that between 1 h time steps there is a high enough resolution so that parcels are advected out of the pixels they currently occupy. If the algorithm was run at $1 \times 1°$, large discretisation errors would be introduced as air would very rarely travel out of the pixel that it occupies within a single hour-long time step. For this reason, the resolution of the DCC dataset is increased to $0.1 \times 0.1°$ using a nearest-neighbour interpolation to be compatible with the TSC algorithm. The resolution of the outputted TSC files is averaged to $1 \times 1°$ to be compatible with the MODIS and CERES gridded products.

The TSC array is interpolated into missing regions after each advection time step (Fig. 2). When each $0.1 \times 0.1°$ pixel in the TSC array is advected forward, some of the trajectories converge to occupy the same pixel, necessarily leaving some pixels without a TSC value. When this occurs, the missing values are interpolated as an average of TSC values around this empty pixel. Between each time step, divergence leaves approximately 5 % of the pixels empty. When two trajecto-

ries enter the same pixel, they will proceed to follow the same trajectory ad infinitum due to the deterministic nature of the trajectories. Therefore the trajectory with the smaller TSC value is the one that determines the TSC value at that pixel from that point on. This increases confidence that any high TSC value really represents air at such long timescales since convection.

The TSC map is defined between 32° S and 32° N to ensure that air is allowed to briefly leave the region of analysis and re-enter. If the wind field brings air from outside 32° S–32° N into this region, this new air will not be assigned a TSC value until it meets new convection or advection introduces a TSC value. This increases confidence in the high-TSC regions. If TSC is defined well beyond the tropics, the extratropics act like a reservoir of very high TSC values that are occasionally advected back into the tropics. This inflates the TSC values because no convection is defined outside of 30° S–30° N.

As TSC is initially undefined, there is an associated spin-up time until every point is a true reflection of the hours since the air at that location last experienced convection. This spin-up time is shown in Fig. 3 and is approximately 20 d. Therefore in all further analysis, the first month of data is excluded.

## 2.4  Detrained versus in situ cirrus

Not all of the cirrus along the advected air parcels will have detrained directly from the convective cores. Therefore it is useful to have an indicator for what percentage of the cirrus are detrained at any one time since convection. To do this a "cirrus type" value is assigned to each grid box where detrained cirrus are present. Initially, all cirrus are considered to be detrained. Once the ISCCP cloud fraction drops below 10 % along a given trajectory, any cirrus that form are considered to be in situ in origin. This cirrus type value does not necessitate that a cirrus cloud be present in the grid box, just that the cloud fraction of the detrained cirrus has dropped below 10 % at some point, meaning that any subsequent cirrus that form are in situ. This is a similar method to that used by Luo and Rossow (2004), who use a threshold of 20 % of the maximum cirrus cloud fraction along a trajectory to identify the "zero detrained cirrus" case, rather than a 10 % cloud fraction threshold as is used in this work.

## 2.5  Evaluation of TSC

After the spin-up, there is minimal variability in the TSC across 2 years of data, with the mean TSC oscillating around 180 h (Fig. 3). The 10th percentile is very consistent, due to the regularity of new convection occurring in the dataset. The 90th percentile is more erratic, oscillating between 350 h and occasionally over 500 h, due to the sporadic nature of the occurrence of and the long tail on the distribution of TSC.

It is interesting to note that the 90th-percentile TSC values are considerably higher than the temporal length of the tra-

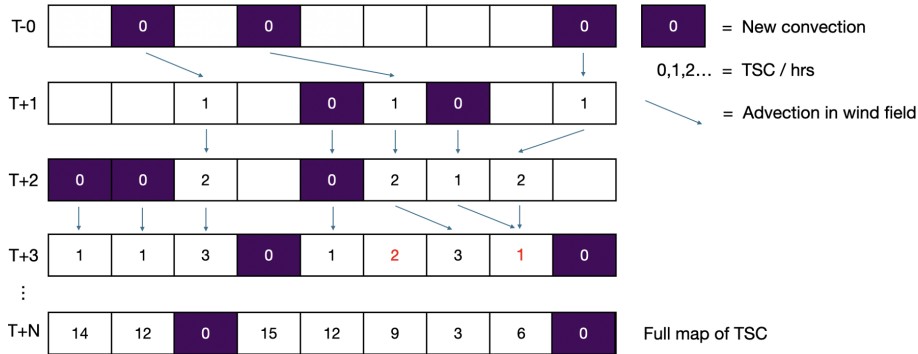

**Figure 2.** Schematic diagram of the advection process in one dimension, assuming all locations start with no TSC value. Red values indicate where either a divergence or a convergence of pixels occurs. The purple boxes show new convections occurring, and the arrows highlight the advection of convective air into new grid boxes between time steps. The values of the TSC are shown in each box.

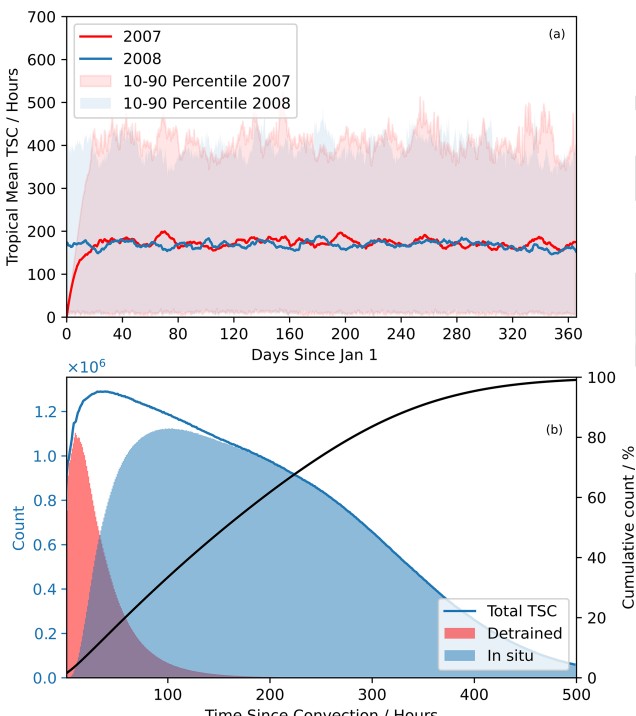

**Figure 3.** **(a)** Tropical mean time since convection (TSC) for 2007 and 2008, with 10th–90th percentiles highlighted. There is little variation in the mean TSC between the years after the initial spin-up period in 2007. **(b)** Histogram of TSC values between 2007 and 2010 and between 0 and 500 h. Red indicates the fraction of grid boxes that contain detrained cirrus; blue indicates candidates for in situ cirrus formation. The right axis shows the cumulative distribution in percent of all TSC values. Of TSC values, > 99 % are less than 500 h.

jectories in prior works (Gehlot and Quaas, 2012; Luo and Rossow, 2004). Both of these studies limited their analysis to 120 h (5 d), which included only 30 % of tropical locations (Fig. 3b). This is a benefit of the TSC approach, which allows the trajectories to be computationally efficiently advected without an upper bound to their length and so does not bias the results towards shorter trajectories and allows for the investigation of clouds that form along the air parcels from convection after the detrained cirrus have dissipated.

Figure 3b shows that there is a sharp decrease in the number of grid boxes containing detrained cirrus clouds after an initial peak at around 12 h from convection. This is in agreement with previous studies that investigated detrained cirrus lifetimes (Luo and Rossow, 2004). Although there are still some detrained cirrus at very long TSC values (more than 100 h), most of the cirrus forming at this point are not directly detrained from convection.

## 2.6 Comparison with HYSPLIT trajectories

A comparison can be made to individual trajectories initiated from locations of convection. The HYSPLIT Lagrangian forward trajectory model is used in this paper (Rolph et al., 2017). The HYSPLIT model uses reanalysis data from the National Centers for Environmental Prediction (NCEP) and the Nation Center for Atmospheric Research (NCEP/NCAR) to advect individual air parcels in a 3D wind field, which includes vertical transport of the air parcel. In this work, HYSPLIT trajectories are initiated from the centre of each $1 \times 1°$ box that contains a TSC-0 value, at an altitude of 10 km. This altitude is approximately 300 hPa in the tropics. These trajectories are followed for 315 h, with the TSC time at the location of each trajectory compared to the time since the HYSPLIT trajectories were initiated. This is done for 3 complete days across 2008 (1 January, 1 June, and 1 September). Figure 4 shows the median value of the TSC for the grid box occupied by the trajectories. The dotted red line indicates all the trajectories initiated from the convective cores (1033 in

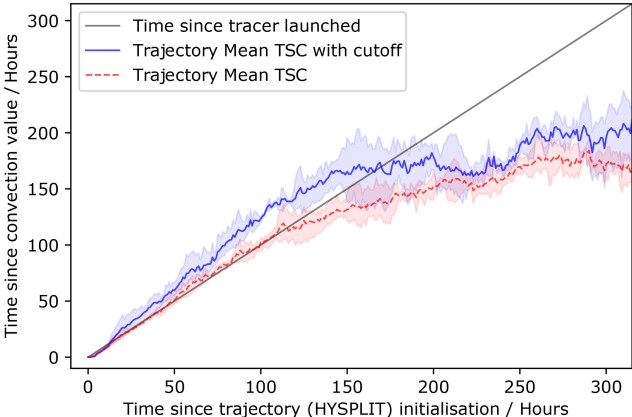

**Figure 4.** Median TSC values (*y* axis) along HYSPLIT trajectories (*x* axis) for all trajectories and for trajectories that are stopped once they reach new convection. The error range is between maximum and minimum trajectory values. The straight line indicates an exact relationship between trajectory age and TSC value.

total), ignoring any new convection along that trajectory. The solid blue line only includes trajectories up to the point at which they next experience convection, which better represents the TSC methodology.

The TSC value and HYSPLIT trajectory time broadly match for lower TSC values (Fig. 4). However, particularly for the trajectories that stop when they experience new convection, the TSC values are higher than the time since the trajectories are initiated. The main reason for this difference is that there is no vertical transport in the TSC advection algorithm. Rather, the average wind speeds (between 200 and 300 hPa) are used to advect the air parcels. The trajectories in the NCEP/NCAR HYSPLIT model are initiated at an average pressure of 200 hPa. This height is an approximation of the DCC detrainment altitude, which leads to some uncertainty in the relevant wind level for advection. During the early stages of the HYSPLIT trajectory, the HYSPLIT wind speed will be greater than that used for the TSC trajectory. This is because the HYSPLIT trajectory is at 200 hPa and the TSC trajectory wind speed is averaged between the 200 and 300 hPa wind speeds.

This explains why the TSC is greater than the age of the trajectories, as the faster HYSPLIT winds will move further from convection than modelled by the TSC dataset. As the trajectories subside (it is worth noting that there is uncertainty in the appropriate subsidence rate), the horizontal wind speeds decrease and start to become closer to the wind speeds in the TSC dataset; therefore the difference between the age of the trajectories and the TSC becomes constant. As the trajectories begin to subside even more, the TSC values become closer to the age of the trajectories, until at around 150 h when the TSC value becomes lower than the age of the trajectories. This close comparison between the TSC and HYSPLIT trajectory times justifies the use of the 200–300 hPa

pressure levels rather than the broader range of pressure levels used by Luo and Rossow (2004).

## 2.7 Cloud properties

The ISCCP-H dataset is used to investigate how the CTP (cloud-top pressure) and $\tau_c$ change as a function of TSC, making use of the joint histograms of CTP–$\tau_c$ in the ISCCP product. These show cloud fraction for a range of CTP and $\tau_c$ bins, produced at the pixel level and limited to show the evolution of the tropical grid box mean values for the region between 30° S and 30° N. The joint histograms are used to isolate the highest clouds, retaining only the top three CTP bins of the histogram (> 375 hPa). This is done by multiplying the centroid value of the top three CTP bins by the cloud fraction of each bin from the joint histograms. This allows for a comparison of the evolution of the $\tau_c$ and CTP for all clouds and just the highest clouds in the tropics.

Failed retrievals in the ISCCP-H dataset are assigned to the lowest $\tau_c$ and CTP bin in the joint histogram. The bin containing these missing values significantly affects the high-cloud $\tau_c$ and CTP, so it is removed when generating the $\tau_c$ and CTP histograms in Fig. 7.

The vertical profile of the cirrus is investigated using the DARDAR dataset, a combination of the CloudSat radar and CALIPSO lidar (Delanoë and Hogan, 2008a). The overpass locations of DARDAR are matched within the hour to the TSC at that location, at $1 \times 1°$ resolution. A $1 \times 1°$ grid box can contain many DARDAR retrievals; each of these DARDAR retrievals is assigned the same TSC value – that of the $1 \times 1°$ grid box. These DARDAR retrievals then all contribute equally to the analysis in the relevant TSC bin. The DARDAR cloud mask is used to filter out aerosol, ground, and unknown retrievals. The DARDAR-Nice product (Sourdeval et al., 2018) is used to provide the vertical profile of ice crystal number concentration ($N_i$) for crystals larger than 5 µm ($N_i^5$) and 100 µm ($N_i^{100}$).

The radiative evolution of convection, from thick convective core to thin anvil cirrus, is investigated by analysing the CERES SYN1deg L3 LW and SW TOA fluxes (NASA/LARC/SD/ASDC, 2017). The CRE is calculated as the all-sky minus clear-sky radiative fluxes from the CERES data. In order to only look at the radiative evolution of the highest clouds, regions where the cloud fraction of the low clouds (as defined by the ISCCP joint histograms) is less than 1 % are classified as "high-cloud-only" grid boxes. For the cloudiest regions in the tropics (which are also the regions of the lowest TSC), there is often no hourly outgoing clear-sky LW data. For these grid boxes, the 3-monthly seasonal average outgoing clear-sky LW data are used in order to calculate a value for the CRE.

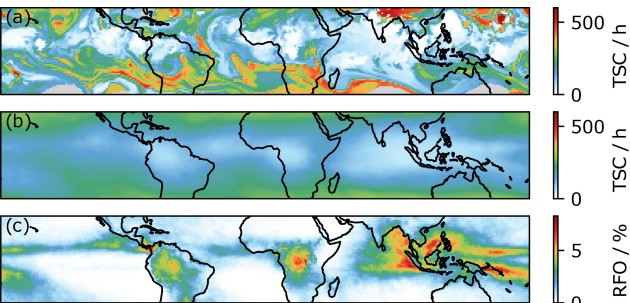

**Figure 5. (a)** Snapshot of the TSC 30° S–30° N at 00:30 UTC on 26 July 2008. **(b)** Mean TSC from between 2008 and 2010. **(c)** RFO of deep convection between 2008 and 2010 (as in Fig. 1).

## 3 Results

### 3.1 Regional distribution of time since convection

Figure 5 shows the map of time since convection for the tropical region 30° S–30° N at 00:30 UTC on 16 July 2008. As might be expected, the TSC map broadly follows the geographical distribution of the deep convective cores shown in Fig. 1. The areas of low TSC, such as the Maritime Continent, South America, and central Africa, align with the regions of high DCC occurrence. However, the low-TSC regions are more spread out around the regions of DCC occurrence, as they are advected via the wind field away from the initial DCCs. Regions of particularly high TSC are the extratropical regions around 30° S and 30° N latitudes, far from convection.

### 3.2 Evolution of ISCCP cloud properties

As the optically thick deep convective cores dissipate, previous studies (Luo and Rossow, 2004) indicate a reduction in the cloud optical thickness ($\tau_c$) as the anvil cirrus decay into thin cirrus and eventually disappear completely. Changes in the cloud-top pressure (CTP) are also expected, moving from low pressure (high altitude) close to the convective core to higher pressures (lower altitude) as the anvil cirrus extends and sinks (Luo and Rossow, 2004).

Figure 6 shows the ISCCP-H joint histograms in the first 5 d since convection, with each 24 h period showing the joint histogram of CTP and $\tau_c$. At day 0 ($0 < \text{TSC} < 24$ h) the histogram is dominated by optically thick, high-altitude clouds with high cloud fractions. There is very little low cloud, which may be due to the thick high clouds obscuring the clouds below them. However, low clouds become visible in the first 24 h after convection. There is a shift to optically thin clouds as TSC increases. This implies that the thick anvil cirrus clouds do not persist longer than 24 h post-convection, consistent with previous studies by Luo and Rossow (2004). This is supported by Fig. 6, which shows the difference is larger between day 0 and day 1 than any subsequent day.

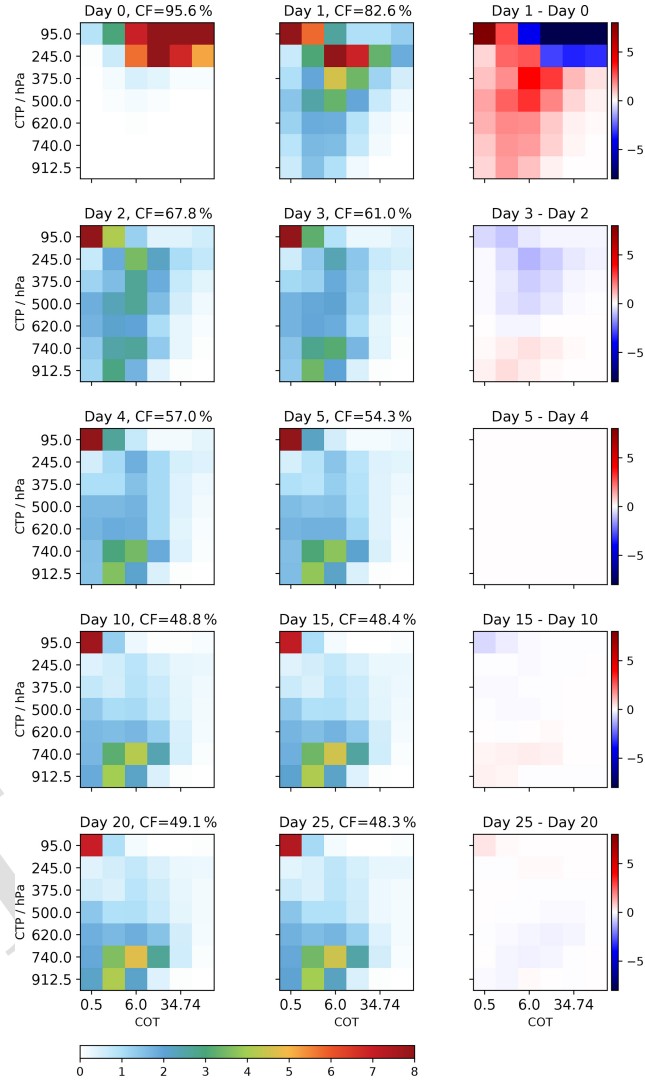

**Figure 6.** ISCCP-H joint histograms showing the distribution of the CTP and $\tau_c$ and cloud fraction as a function of days since convection. The rightmost column shows the absolute percentage change between days. Day 0 is the first 0–24 h since convection, day 1 is 24–48 h since convection, and so on. High CF values for the upper right grid boxes would indicate a distribution skewed towards optically thick, high-altitude clouds and the opposite (optically thin, low-altitude clouds) for bottom left grid boxes.

High clouds with an optical thickness in the middle two bins (between 2.3 and 4.5), representing the thicker cirrus outflow, still noticeably persist up to day 5, with the most significant decrease in their cloud fraction happening 3 d after convection. They remain with low cloud fractions up to day 25.

From day 4 onwards, low clouds begin to dominate in the joint histogram; however there is still a presence of thin cirrus. Interestingly, small changes in the cloud fractions are still seen between day 10 and day 15, particularly an increase in low cloud and decrease in high cloud. This implies that the

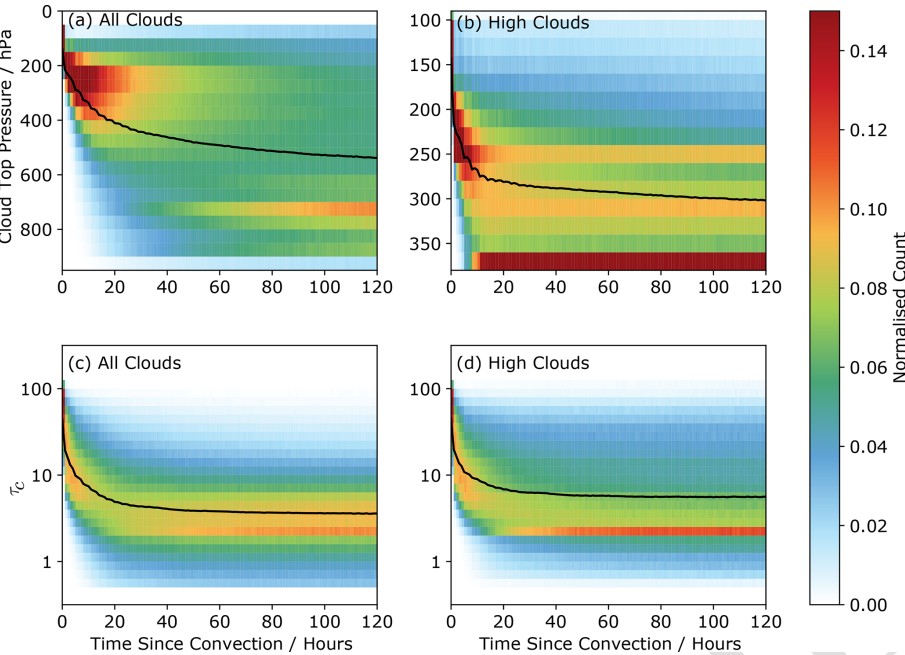

**Figure 7.** ISCCP cloud properties as a function of TSC. The normalised distribution represents the probability of finding clouds with a particular CTP or $\tau_c$ for a given TSC value. The black line indicates the mean value. The CTP and $\tau_c$ **(a, c)** for all clouds and **(b, d)** for high clouds only.

TSC is still influential at up to 360 h from convection. Note that the top left bin in each histogram in Fig. 6 (high, optically thin cloud) remains particularly frequent across all histograms. This is due to the failed retrievals. These cannot be assigned to a particular $\tau_c$/CTP bin and so are assigned into the top left bin in the ISCCP-H dataset by default (Pincus et al., 2012), as mentioned in Sect. 2.6.

These results are generally consistent with previous work (Gehlot and Quaas, 2012; Luo and Rossow, 2004); however this paper sees a slightly quicker decay of the anvil cirrus than in Luo and Rossow (2004) and also sees continued changes in cloud fraction beyond 5 d from convection.

Luo and Rossow (2004) use a column-averaged wind field between 200 and 500 hPa, whereas this work uses 200–300 hPa to better reflect the range of heights at which the detrained thin cirrus and in situ cirrus occur. The wind field in this work will be on average faster than Luo and Rossow (2004), and therefore the air will travel further in the same period of time. As such, a TSC value of 10 h will be geographically further along the trajectory than in Luo and Rossow (2004). This will cause the lifetime of the convective outflow clouds to appear shorter and may explain these differences from their work.

Seeing changes in the cloud fraction at such long TSC values is a consequence of the approach of this work. Unlike previous studies where the trajectories were stopped at predetermined times, this work tracks air from convection for an unbounded amount of time. Therefore changes in cloud properties can be tracked at much longer times since convec-

tion than in previous work, without having to pick out specific convective events that are known to reach a certain age. The TSC methodology also explicitly deals with scenarios in which a trajectory is overlapped by new convection.

## 3.3 Evolution of cloud properties

Figure 7 shows the evolution of the CTP and $\tau_c$ respectively as a function of the TSC for both all-cloud grid boxes (Fig. 7a, c) and the high-cloud-only grid boxes (Fig. 7b, d). The joint histograms are normalised so that the number of $y$ axis bins adds up to unity to account for the fact that there are more retrievals for smaller TSC values. This provides a higher temporal resolution than the histograms in Fig. 6.

Figure 7a shows that the CTP initially sits at very high altitudes for the convective cores, around 200 hPa, coincident with the upper-level wind field used. There is a large increase in the mean CTP in the first day since convection as the convective core thins, and the mean CTP is skewed by the optically thicker, lower-altitude clouds (shown in Fig. 6). The histogram still shows a presence of higher-altitude clouds, with an increasing proportion of low-altitude clouds further from TSC-0 (the convection itself). There is still a decreasing CTP at 5 d from convection.

Isolating just the high clouds using the ISCCP histograms, Fig. 7b shows a similar decline in the mean CTP, albeit at lower pressures. The $\tau_c$ also sits at 200 hPa initially, decaying to 250 hPa in the first 6–12 h as the convective core dissipates. The decay then slows, reaching 300 hPa at 120 h from

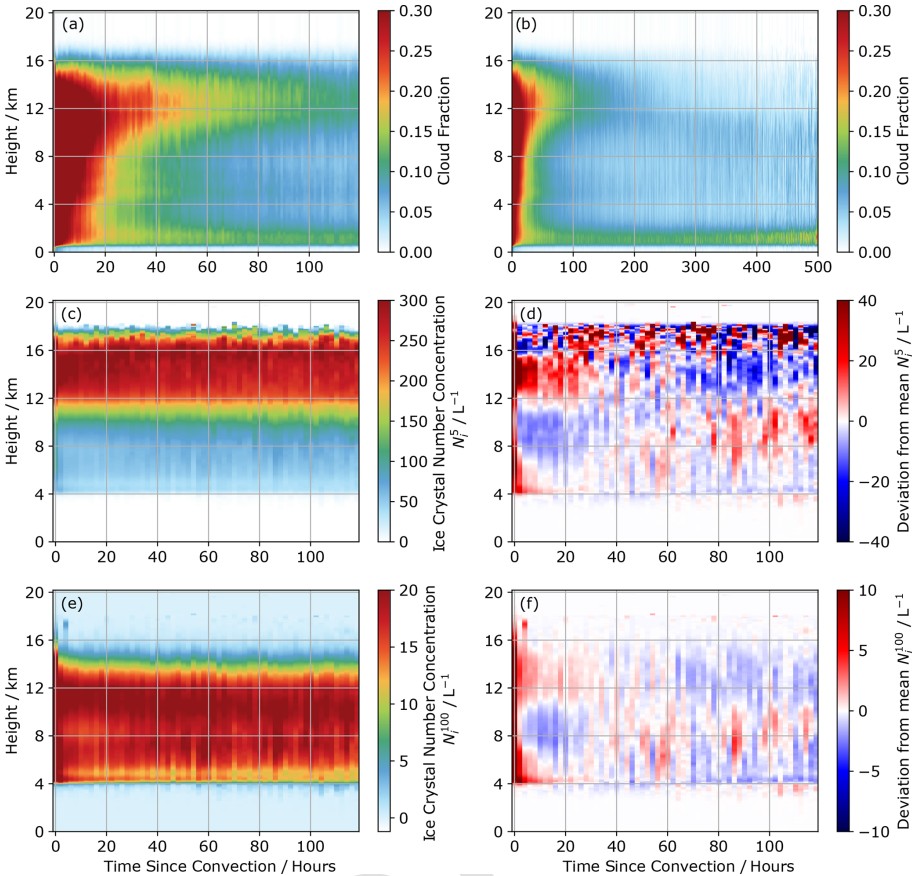

**Figure 8.** Vertical evolution of DARDAR cloud properties as a function of TSC. **(a)** Cloud fraction up to 120 h from convection. **(b)** Cloud fraction up to 500 h from convection. **(c)** $N_i$ for particles larger than 5 µm. **(d)** $N_i^5$ anomaly for 120 h from convection. **(e)** $N_i$ for particles larger than 100 µm. **(f)** $N_i^{100}$ anomaly from mean value for 120 h from convection. Anomaly here refers to the difference in the $N_i$ at a given TSC from the mean $N_i$ at that altitude for the tropics.

convection. There is a significant cloud layer at 370 hPa that becomes visible as the high cloud dissipates. These are likely clouds not directly detrained from convection that become visible as the cloud fraction of the anvil cloud above decreases, and any changes in the properties of these clouds, particularly at large TSC values, are not suggested to be directly related to the initial convection.

Figure 7c shows the evolution of the cloud optical thickness for all clouds. The all-cloud $\tau_c$ drops significantly in the first 24 h from convection, indicating the dissipation of the very thickest high clouds associated with the convective cores. This is to be expected as the regime shifts from the thicker cirrus associated with convection to thin cirrus that either persists after a convective event or is formed in situ and unrelated to convection.

When only looking at high clouds in Fig. 7d, the mean values of the histogram are relatively unchanged. However there is larger spread in the cloud optical thicknesses, including a relative increase in the optically thinnest of the clouds in the lowest bin.

## 3.4 Vertical profile evolution

The vertical distribution of cloud and $N_i$ varies strongly as a function of TSC. Figure 8a shows the cloud amount and thus the general anvil shape of convective cores and their associated cirrus outflow up to 120 h from convection. Figure 8b shows the same up to 500 h from convection. At very low TSC (TSC < 2 h), there is a substantial amount of cloud at all altitudes up to approximately 16 km associated with the very tops of the convective towers. It should be noted that this cloud amount is not comparable to the ISCCP cloud fraction used to identify the deep convection. The low- to mid-altitude cloud, likely the convective cores, quickly dissipates as the TSC increases. High clouds are maintained for much longer, with changes in the high cloud fraction continuing beyond 120 h from convection. There is a moderate quantity of low clouds, almost independent of TSC. There is also a sustained quantity of mid-level clouds between 4 and 8 km for the first 60 h. These mid-level clouds are thought to form when the latent heat associated with convective freezing creates a shallow temperature inversion, which slows vertical motion in

subsequent convective cycles and detrains moisture near the freezing level (Bourgeois et al., 2016). Figure 8b shows that even at very long TSC values, from 120 up to 500 h, there is still a significant change in cloud amount as a function of TSC.

Figure 8c shows the ice crystal number concentration for particles greater than 5 μm in diameter ($N_i^5$) as a function of the time since convection. Figure 8e shows the same as Fig. 8c but for those greater than 100 μm ($N_i^{100}$). Given that the $N_i$ is a very strong function of temperature (and therefore altitude), it is useful to consider the relative changes in the $N_i$ as a function of TSC. Figure 8d and f show the deviation of the $N_i$ from the tropical mean $N_i$ for a given altitude. Red indicates where the $N_i$ at a given TSC is greater than the tropical mean, and blue indicates those regions where it is less than the tropical mean.

Considering Fig. 8d first, there is a greater-than-average $N_i^5$ at all altitudes close to convection. This is most notable between 12 and 16 km, where the $N_i^5$ peaks. This is to be expected given the presence of a deep convective core. At short times after TSC-0, there remains an above-average $N_i^5$ between 12 and 16 km. This is indicative of the convective outflow that remains after the convective core has dissipated. This outflow decays within 36–42 h, when the deviation to the mean $N_i^5$ becomes negative. There is also an increase in $N_i^5$ at similar times at lower altitudes (between 4 and 8 km) that decays within around 20 h. This is likely due to the active shallow convection that occurs close to the region of deep convection and thin mid-level clouds discussed in Bourgeois et al. (2016). Considering Fig. 8f, there is a clear increase in the $N_i^{100}$ close to convection, particularly at lower altitudes between 4 and 8 km. Although the DARDAR retrieval is less reliable in convective cores (Delanoë and Hogan, 2008a), larger ice crystals are expected in the convective core at lower altitudes as they are less likely to be lifted higher into the cloud column than smaller ice crystals, and they also subside quicker (Jensen et al., 2018). The lower-altitude $N_i^{100}$ drops off rapidly as the convective core dissipates. As in the case of the $N_i^5$, there is an increase in the $N_i^{100}$ between 12 and 16 km, which is sustained for the first 24 h from convection and linked to updraughts of large ice crystals from deep convective cores. In both of Fig. 8d and f, there appears to be anomalies after 36–42 h from clouds not directly detrained from convection. This is expected, as not all high clouds are detrained from convection or change as a function of TSC; as shown in Fig. 3b, many of the cirrus clouds that form along the trajectories form in situ after the detrained cirrus have dissipated.

## 3.5   TSC as a function of latitude

It is clear from Fig. 5b that on average TSC increases with latitude, as is expected from the large-scale behaviour of the Hadley circulation. However, while latitude can explain the broad features of the high cloud field, TSC offers valuable

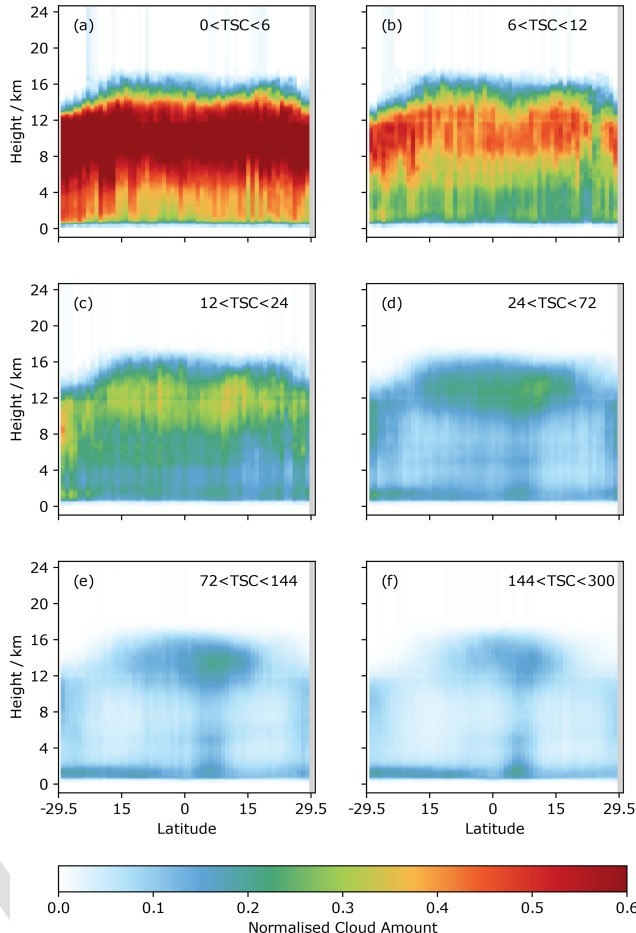

**Figure 9.** Zonally averaged DARDAR vertical cloud amount for a given TSC bin. TSC bins are given in the top right corner.

additional insight into the development of high clouds by more closely constraining their properties and development.

Figure 9a shows the zonally averaged DARDAR cloud fraction in the tropics for all grid boxes with a TSC value between 0 and 6 h. Close to convection, it is clear there is very little latitudinal dependence in the cloud profile when controlling for TSC. As TSC increases during the first 24 h from convection (Fig. 9b, c), latitude starts to become relevant for determining the cloud profile, although the impact is still small. At 3 d from convection (Fig. 9d), the zonal cloud fraction begins to resemble the average tropical climatology. Despite this, there are still clear changes in cloud fraction as a function of TSC at a given latitude, even several days after convection (Fig. 9e, f).

This demonstrates that the results shown in Fig. 8 are not purely due to the correlation between TSC and latitude. In contrast, TSC provides significant additional information about the state of the cloud field. In all regions of the tropics, the temporal evolution of clouds following convection is critical to understanding high-cloud properties.

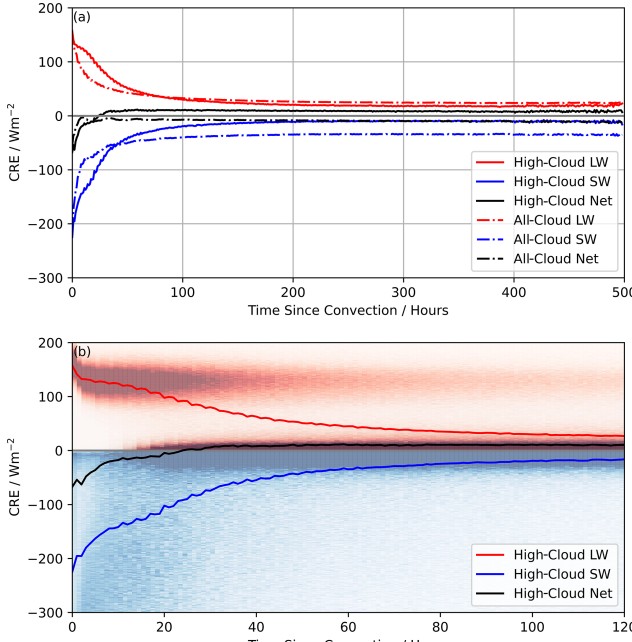

**Figure 10.** Cloud radiative effect (CRE) as a function of TSC. **(a)** All-cloud and high-cloud LW, SW, and net CRE up to 500 h from convection. **(b)** High-cloud LW, SW, and net CRE up to 120 h from convection with normalised joint histogram showing the probability of finding a particular CRE value (LW in red, SW in blue) for a given TSC value.

## 3.6 Cloud radiative effects

The radiative evolution of convection, from thick convective core to thin anvil cirrus, is characterised by analysing the CERES SYN1deg L3 LW and SW TOA fluxes (NASA/LARC/SD/ASDC, 2017). Figure 10 shows the SW, LW, and net cloud radiative effect (CRE) as a function of the time since convection. Figure 10a shows the evolution of the LW, SW, and net CRE for both the all-cloud grid boxes and grid boxes containing only high cloud, up to 500 h from convection.

Figure 10a shows that when considering the all-cloud CRE, the net CRE is always slightly cooling, with an oscillation in the first 48 h due to the diurnal cycle in the CRE. This net CRE of close to zero in the tropics is well studied (Ramanathan et al., 1989; Hartmann and Berry, 2017; Wielicki et al., 1996). In the all-cloud case, there is initially a very large, significant warming and cooling that cancel each other out. The LW and SW CRE decrease significantly in the first 24–48 h as the cloud fraction of these clouds drops (Fig. 8a). By 100 h from convection there is very little change in the all-cloud and high-cloud CRE. To understand how the radiative forcing of the clouds that evolve from convection changes, it is vital to consider only the grid boxes containing high clouds.

The high-cloud CRE is a strong function of TSC at very low values of TSC (Fig. 10b). Initially, there is a large-SW

and large-LW component to the CRE, with $-210\,\mathrm{W\,m^{-2}}$ TS1 SW cooling and $150\,\mathrm{W\,m^{-2}}$ LW warming at TSC-0 leading to a net CRE of $-70\,\mathrm{W\,m^{-2}}$ TS2 cooling. This is due to the very optically thick convective core that reflects a significant portion of the incoming solar radiation (Fig. 7d) whilst also having a very large LW warming due to its high cloud tops (Fig. 7b).

As the convective core dissipates and the cloud optical thickness decreases, the SW CRE drops significantly, to $\sim 100\,\mathrm{W\,m^{-2}}$ TS3 at $\sim 20\,\mathrm{h}$ TS4 from convection, whereas the LW warming remains at $\sim 90\,\mathrm{W\,m^{-2}}$. At 24 h from convection, the net CRE shifts from a cooling regime to a warming regime, dominated by the LW warming of the thinner detrained cirrus in the initial stages of TSC and then sustained by in situ cirrus formed along the trajectories. After the initial dissipation of the core, the net warming is sustained. After 24 h since convection, the net CRE remains at $\sim 10\,\mathrm{W\,m^{-2}}$ TS5. The histogram in Fig. 10b shows the LW warming splitting into two layers, with the weighted mean value falling in between. The top layer, which sits at around $100\,\mathrm{W\,m^{-1}}$, is the high cirrus with a high cloud fraction that has the largest warming potential. The layer that emerges close to $0\,\mathrm{W\,m^{-2}}$ is due to the lower-cloud-fraction high clouds that emerge further from convection. These clouds may not be directly detrained from convection themselves but still evolve as a function of TSC. With their low cloud fraction, their CRE is small, close to $0\,\mathrm{W\,m^{-2}}$ in both the LW and SW.

This sustained warming as a function of TSC is not seen in previous studies. Gehlot and Quaas (2012) found the net CRE in the ECHAM5 model to be negative up to 5 d (120 h) from convection. However they did not isolate the high clouds, meaning that cooling from lower-altitude thick clouds contributed to their CRE. Without removing low clouds from the analysis, the work in this paper finds a similar sustained, albeit small, cooling for the all-cloud CRE (Fig. 10).

Our results differ to previous model studies. Gasparini et al. (2021) looked at the CRE evolution in E3SM, a high-resolution GCM. These results found that, after an initial positive CRE due to small insolation values where convection occurred early before sunrise, the SW and LW CRE quickly decayed, and the net CRE became negative after 2 h, oscillating to a small warming again at 15 h as the solar insolation decreased. These simulations did not represent the thinnest cirrus particularly well, so they may underestimate some of the LW warming seen in this work.

## 4 Discussion

There are multiple benefits to the TSC approach described in this paper. It is computationally efficient when compared to tracking every single location of deep convection in the tropics. Instead of keeping and updating an array of locations for each new convective event, which grows with each

time step, this work has just one TSC array for the entirety of the tropics, which is updated at each time step. It also removes any bias in selecting specific initial convection, since all stages of the convection from the shortest lived to the longest lived contribute to the mean TSC. The unbounded nature of the tracking means that the full lifecycle of all trajectories from convection are considered until they are superseded by new convection. Finally, detrained cirrus can be differentiated from in situ cirrus. The majority of the detrained cirrus dissipate after around 12–24 h from convection; however some last as long as 100 h from convection. The in situ cirrus that form along trajectories still change as a function of TSC, and the formation mechanisms responsible for the development of in situ cirrus may be affected by the deep convection that has lifted air and moisture up to the tropopause, even after the detrained cirrus has dissipated.

However, there are some uncertainties introduced in the TSC algorithm. Firstly, it only considers horizontal wind fields, averaged between 200 and 300 hPa, and there is no vertical advection of the TSC. This is a similar assumption to that made by Luo and Rossow (2004), except that they considered a larger pressure-averaged wind field between 200 and 500 hPa. The choice of the 200–300 hPa average wind field is supported by the evolution of the ISCCP CTP (Fig. 7) and the HYSPLIT trajectory pressures. There is some vertical subsidence during the evolution of the cirrus; therefore using the 200–300 hPa wind field means there is some uncertainty in the advection of the air from deep convection. Adding a vertical evolution component would be an option for future enhancement to the TSC algorithm. It would add considerable computational complexity but would improve the calculations of TSC and reduce the uncertainty introduced by using a pressure-averaged wind field.

Secondly, there is necessary interpolation that occurs when the wind field diverges. If a grid box becomes empty between time steps, then the value for the TSC in that grid box is calculated as the mean value of the surrounding grid boxes. This increases the uncertainty in longer TSC values. Two different interpolation methods have been tested, with minimal impact on the represented results, suggesting this is not a large source of uncertainty (see Supplement).

While there are significant changes in DARDAR ice properties as a function of TSC (Fig. 8), there are still considerable uncertainties surrounding the retrieval, particularly at warmer temperatures. The DARDAR-Nice $N_i$ retrieval compares favourably to in situ measurements at the cirrus temperatures that are the focus of this work (Sourdeval et al., 2018). However, the potential temporal variation of factors such as particle shape might introduce TSC-dependent biases in the $N_i$ retrieval, producing an apparent change in $N_i$. This will be investigated in future work.

Two key factors may introduce temporal biases in the CRE presented in Fig. 10. First, to ensure CRE values existed for all clouds, missing LW clear-sky values in the instantaneous data were filled with the 3-month seasonal mean. This intro-

duces some uncertainty into the results but using annual averages makes little difference to the results. Using too short an averaging period results in large amounts of missing data near convection.

Furthermore, it is important to note that the ISCCP dataset does not necessarily provide a true representation of cloud-top pressure or cloud optical depth. As stated in Chen and Del Genio (2009), the ISCCP CTP-TAU histograms are not actual vertical distributions of clouds. This means that when considering Fig. 6, the apparent cloud layer that appears at 370 hPa should not be taken to be a true appearance of the clouds at this height. More likely, it is the existence of thin high clouds sitting over low cloud that causes ISCCP to incorrectly diagnose them as lower-level clouds (Mace and Wrenn, 2013). This occurs when the optically thick deep convection dissipates, and the thin cirrus moves over a region of continuous low cloud in the trade cumulus regions. The DARDAR dataset and Fig. 8 are particularly useful here as they can provide us with a more reliable vertical distribution of the cloud amount. The issue with the ISCCP mid-level clouds is not concerning and should not impact our subsequent results, in particular the CRE calculations, in Fig. 10. To isolate our high-cloud regions for the CRE calculations we select bins with very few low clouds, making it unlikely that this mid-level cloud effect drives the results in Fig. 10.

## 5  Conclusion

The lifecycle of convective cores and their associated cirrus outflows, including their vertical evolution, have been characterised in this paper through the creation of a "time since convection (TSC)" dataset. Convection is identified in satellite data (Fig. 1) and then advected as a tracer in a reanalysis wind field, resulting in every point in the tropics being assigned a TSC value that indicates the last time that air parcel experienced convection (Fig. 4). Building on previous studies that were forced to set an artificial limit to the time they tracked a parcel of air from convection (Luo and Rossow, 2004; Gehlot and Quaas, 2012), the methodology presented here allows an effectively unbounded upper limit for the time since convection. Changes in high clouds as a function of TSC continue to be observed several days after the convective event (Fig. 9).

Cloud properties are strong functions of TSC, particularly those for high clouds. There is a general trend for high-altitude, optically thinner clouds to appear over time, as expected in the deep convection to anvil to thin cirrus evolution (Fig. 7). Following a sharp initial decay in both the CTP and COT in the first 12 h from convection, a slower decrease is observed in ISCCP data over the next 120 h. This gradual thinning of the anvil is also seen in vertical profiles of cloudiness, with the clearest changes observed in the first 120 h (Fig. 8a), although changes in cloud fraction as a function of TSC continue to be observed for several hundred hours

after the convective event (Fig. 8b) in both detrained and in situ cirrus.

Cloud properties and TSC are both strong functions of latitude in the tropics (Fig. 4). However, changes in the cloud properties along TSC trajectories are not solely due to latitudinal changes along these trajectories. Controlling for latitude by looking at the zonally averaged cloud amount at a given TSC, it is clear that cloud properties are similar for a given TSC at all latitudes. This shows that looking at evolution along trajectories is not merely characterising the changing climatology as trajectories move from the tropics to extratropics, but it also provides significant extra information on cloud evolution, making it a powerful tool for studying the development of clouds.

The changes in cloud properties explain the observed evolution of the CRE as a function of TSC (Fig. 10). Considering the CRE for all clouds, the net CRE is very close to zero at all values of TSC. This apparent balance in the tropical CRE is well documented (Wielicki et al., 1996; Ramanathan et al., 1989; Hartmann and Berry, 2017), occurring both close to convection when the LW and SW values are very high and at very large TSC values. However, when the high-cloud-only regions are isolated, there is a sustained warming beyond 120 h from convection after a brief initial cooling. The warming effect of high clouds is only unmasked when removing the optically thick low clouds, which generate a significant SW cooling at large TSC values (Fig. 7), from the analysis. The LW warming from these high, optically thin clouds is sustained well beyond 24 h from convection, by which point the SW cooling from the convective core has largely dissipated. With this close CRE balance at a range of TSC values, understanding the controls on high-cloud evolution as a function of TSC is essential to constrain the processes governing tropical radiation balance.

This paper has introduced a novel method for assessing the evolution of clouds following convection, through anvil cirrus to thin cirrus, by combining satellite and reanalysis data to calculate the "time since convection" for each point in the tropics. This provides a new window into high-cloud development in the tropics. The flexible nature of the TSC approach allows it to be easily applied to a wide range of cloud and atmospheric datasets. Whilst there are significant changes in CRE with TSC observed close to convection, this work also demonstrates that changes in cloud properties with TSC can be observed at timescales up to 500 h from convection, long after the original convective event has dissipated. With previous studies being limited to the youngest 30 % of tropical locations ($< 120$ h), there is a clear need for future investigation of cirrus development at these longer timescales.

**Data availability.** The data used is highlighted in Sect. 2.1. The ISCCP-H data were obtained from the NCEI/NOAA Climate Data Records (https://doi.org/10.7289/V5QZ281S, Rossow et al., 2017). The ERA5 data were obtained from the Climate Data Store (https://doi.org/10.24381/cds.bd0915c6 TS6, Hersbach et al., 2018). The CERES data were obtained from the NASA EARTHDATA ASDC (https://doi.org/10.5067/TERRA+AQUA/CERES/SYN1DEG-1HOUR-L3.004A TS7). The DARDAR dataset was obtained directly from ICARE (Delanoë and Hogan, 2008b).

**Supplement.** The supplement related to this article is available online at: https://doi.org/10.5194/acp-23-1-2023-supplement.

**Author contributions.** Both authors contributed to study design and interpretation of results. GH performed the analysis and prepared the manuscript with comments from EG.

**Competing interests.** The contact author has declared that neither of the authors has any competing interests.

**Disclaimer.** Publisher's note: Copernicus Publications remains neutral with regard to jurisdictional claims made in the text, published maps, institutional affiliations, or any other geographical representation in this paper. While Copernicus Publications makes every effort to include appropriate place names, the final responsibility lies with the authors.

**Acknowledgements.** This work was supported by a Royal Society University Research Fellowship (URF\R1\191602) and a PhD studentship from the Department of Physics, Imperial College London. The authors gratefully acknowledge the NOAA Air Resources Laboratory (ARL) for the provision of the HYSPLIT transport and dispersion model and READY website (https://www.ready.noaa.gov, last access: 12 December 2022) used in this publication. The authors also gratefully acknowledge Odran Sourdeval and Athulya Saiprakash (University of Lille) for providing the DARDAR data and helpful discussions and suggestions. The authors also thank the reviewers and editor for their comments, which greatly improved this work.

**Financial support.** This research has been supported by the Royal Society (grant no. URF\R1\191602).

**Review statement.** This paper was edited by Timothy Garrett and reviewed by Gerald Mace and one anonymous referee.

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

## Remarks from the typesetter

TS1    This correction needs the editors' approval. Please give a description why this value has to be changed.

TS2    This correction needs the editors' approval. Please give a description why this value has to be changed.

TS3    This correction needs the editors' approval. Please give a description why this value has to be changed.

TS4    This correction needs the editors' approval. Please give a description why this value has to be changed.

TS5    This correction needs the editors' approval. Please give a description why this value has to be changed.

TS6    Please confirm change of DOI number.

TS7    Please check DOI number and provide reference list entry.

TS8    Please confirm added citation.