# Peer review of "The evolution of deep convective systems and their associated cirrus outflows"

_Atmospheric Chemistry and Physics, 2022_

## Referee Comment (RC1)

Review of: "The evolution of deep convective systems and their associated cirrus outflows" by George Horner and Edward Gryspeerdt

The authors aim to study the evolution of deep convective systems and their associated cirrus outflows, as the title of the manuscript suggests. As far as I understand, the most significant finding of this work compared to existing literature is the sustained warming effect of convectively detrained cirrus clouds. The authors argue that this effect is sustained past 5 d since the time of the last deep convective event. My main concern is whether the cirrus clouds captured along the trajectories, especially at an increasingly long time since convection (TSC), are actually associated with convection. If the goal of the paper is to evaluate the radiative effect of the clouds associated with convection, the authors would need to properly separate the clouds associated with convection and those that are not. The authors have not provided the analysis to do this.

The main argument that the authors gave to justify that deep convection leads to the formation of the clouds along the trajectories is the expectation of changes in cloud properties with the TSC. Specifically, as the anvil cirrus decays into thin cirrus, it is expected that the cloud fraction decreases, the cloud top pressure increases (as clouds descend to lower altitudes), and the cloud optical depth decreases. These behaviours are seen clearly in the data the first 24 h (Fig. 7), but then the signals become weak (Fig. 6) and/or disconnected in time and/or space (Fig. 7). In Fig. 7(b), high clouds are seen at 250 hPa, 310 hPa, and 370 hPa, but only the clouds at 250 hPa and 310 hPa appear to be connected to the convective anvil, while the clouds at 370 hPa do not. Where do the clouds at 370 hPa come from? It is odd that clouds appear at 370 hPa at TSC = 10 h before the thick convective anvil above dissipates. Is this an artifact of the data and/or the method? If so, is the data and/or method trustworthy? In addition, the cloud top pressures of the high clouds at respectively 250 hPa, 310 hPa, and 370 hPa do not increase with time (Fig. 7b) as expected, and the optical depth of the clouds does not decrease with time either after about 30 h (Figs. 7c and 7d).

Figures 8(d) and 8(f) show that the convective outflow decays within 36 h–42 h (line 285 in the manuscript). Subsequent anomalies in ice number concentration are not discussed in the manuscript, but they appear to me to be disconnected to the convective outflow anvil in these figures. In short, both Figs. 7 and 8 suggest that a large number clouds along the trajectories may not be associated with convection (or at least not directly), particularly for long TSC.

Besides the lack of identification of clouds that originate from convection and those that do not, the other issue I see is the error and uncertainty associated with the trajectory calculation, which increases with TSC. This means that the results for TSC > 5 d may not be meaningful. The error associated with omitting the vertical motions and using the wind averaged between 200 hPa and 300 hPa has been discussed in the manuscript (Section 2.5). However, I am concerned with the error associated with the method of calculating the TSC, in which if a grid box becomes empty, then the value for the TSC in that grid box is taken as the mean value of the surrounding grid boxes (as stated in lines 343–344 in the manuscript). How much is the error that this introduces? In addition, the number of retrievals decreases with TSC. Figure 7 shows the distribution of clouds along the trajectories, but it has been normalized so we do not see the actual number of retrievals. How many retrievals are there for TSC longer than 5 d? Is this number sufficiently large to guarantee a reliable result? I suppose that a detailed analysis of the error and uncertainty as TSC increases is

needed. Given the error and uncertainty, should the trajectory analysis be limited for TSC < 5 d? But if so, what would be the new finding of this work compared to Luo and Rossow (2004)'s? Recall that Luo and Rossow (2004) already performed forward trajectory calculations for 5 d starting from deep convective systems to address a similar topic.

In summary, the major issue of this work is the lack of identification of clouds that originate from convection and those that do not. In addition, the error and uncertainty associated with the trajectory calculation method have not been assessed properly. I hope that the authors will consider these points to improve their work before it can be published in ACP.

**References**

Z. Luo and W. B. Rossow. Characterizing Tropical Cirrus Life Cycle, Evolution, and Interaction with Upper-Tropospheric Water Vapor Using Lagrangian Trajectory Analysis of Satellite Observations. *J. Climate*, 17(23):4541–4563, 2004.

---

## Author Comment (AC1)

**Reviewer 1**

*: The authors aim to study the evolution of deep convective systems and their associated cirrus outflows, as the title of the manuscript suggests. As far as I understand, the most significant finding of this work compared to existing literature is the sustained warming effect of convectively detrained cirrus clouds. The authors argue that this effect is sustained past 5 d since the time of the last deep convective event. My main concern is whether the cirrus clouds captured along the trajectories, especially at an increasingly long time since convection (TSC), are actually associated with convection. If the goal of the paper is to evaluate the radiative effect of the clouds associated with convection, the authors would need to properly separate the clouds associated with convection and those that are not. The authors have not provided the analysis to do this.*

**Reply**: We thank the reviewer for their comments which we address in turn below. The goal of the paper is not to track the evolution of exclusively detrained clouds from convection, as this has been done before in mesoscale studies (e.g. Luo and Rossow, 2004). This study instead investigates the evolution of air parcels (and the associated cloud properties) advected from convective cores. While some of the clouds studied will have detrained from convection, as the reviewers note, this study is not limited to these clouds. In contrast to previous studies, this work uses time since convection (TSC) to characterise all tropical high clouds and their radiative properties, irrespective of whether they were directly detrained from convective cores. We also include new analysis of how TSC varies as a function of latitude. We have modified the text in the introduction, discussion and conclusion to better highlight this difference to previous work.

During the preparation of this response, the method for interpolating between diverging trajectories has been modified to be more consistent with the description in the methods. These changes are discussed in detail on Page 3 of the response, but have a minimal impact on the results presented in this work.

We have also harmonised some of the terminology, using 'pixel' to refer to an element of the 0.1° by 0.1° used for the TSC calculation, where 'gridbox' is an element of the 1° by 1° grid.

**General comments:**

*: The main argument that the authors gave to justify that deep convection leads to the formation of the clouds along the trajectories is the expectation of changes in cloud properties with the TSC. Specifically, as the anvil cirrus decays into thin cirrus, it is expected that the cloud fraction decreases, the cloud top pressure increases (as clouds descend to lower altitudes), and the cloud optical depth decreases. These behaviours are seen clearly in the data the first 24 h (Fig. 7), but then the signals become weak (Fig. 6) and/or disconnected in time and/or space (Fig. 7).*

**Reply**: We agree that the signal becomes weaker at longer times since convection, but suggest that this is expected. As the reviewer notes, the behaviour of clouds at longer times is less likely to be related to the properties of the initial convection. This will be addressed in future work. However, we note that cloud properties still vary as a function of TSC even at very large TSC values. We have included a new section of analysis and extra plot into the paper that investigates the role that latitude plays in the evolution of the cloud properties, and how the influence of TSC changes as air parcels get further from convection (i.e. at large TSC values). These results are particularly important to isolate the direct influence of TSC on the changes in cloud properties along trajectories, and highlights that these changes aren't merely due to latitudinal influence on cloud properties. This begins on line 315 in the section 'TSC as a function of latitude'.
* * *
*: In Fig. 7(b), high clouds are seen at 250 hPa, 310 hPa, and 370 hPa, but only the clouds at 250 hPa and 310 hPa appear to be connected to the convective anvil, while the clouds at 370 hPa do not. Where do the clouds at 370 hPa come from? It is odd that clouds appear at 370 hPa at TSC = 10 h before the thick convective anvil above dissipates. Is this an artifact of the data and/or the method? If so, is the data and/or method trustworthy? In addition, the cloud top pressures of the high clouds at respectively 250 hPa, 310 hPa, and 370 hPa do not increase with time (Fig. 7b) as expected, and the optical depth of the clouds does not decrease with time either after about 30 h (Figs. 7c and 7d).*

**Reply**: It is difficult to make a definite statement about the original of the clouds at 370hPa due to the overlying high cloud - but we note that they become visible as the thick initial layer of cloud at 200-250hPa dissipates. This interpretation is supported by the DARDAR cross section plots (Figs. 8a,b), which do not show the appearance of a new layer at 8/9km (approximately 370hPa). We have modified the text to make it clearer that the these clouds may not be directly related to the initial convection on lines 272-275: *There is a significant cloud layer at 370hPa that becomes visible as the high cloud dissipates. These are likely clouds not associated with convection that become visible as the cloud fraction of the anvil cloud above decreases, and any changes in the properties of these clouds, particularly at large TSC values, are not suggested to be directly related to the initial convection.*
* * *
*: Figures 8(d) and 8(f) show that the convective outflow decays within 36 h–42 h (line 285 in the manuscript). Subsequent anomalies in ice number concentration are not discussed in the manuscript, but they appear to me to be disconnected to the convective outflow anvil in these figures. In short, both Figs. 7 and 8 suggest that a large number clouds along the trajectories may not be associated with convection (or at least not directly), particularly for long TSC.*

**Reply**: A number of clouds along the trajectories may indeed be disconnected from the initial convective event. It is reasonable to assume that the number of clouds unrelated to convection will increase as a function of time since convection. However it is still of scientific value to investigate how cloud properties from clouds not directly detrained from deep convection may change as a function of time since convection, due to the impact of convection on the properties of the UTLS region, including humidity and aerosol environment. As highlighted above, this point has been clarified on lines 272-275, with a discussion section included on line 315 and a plot included that shows how the cloud distribution is independent of latitude for a given TSC value.
* * *
*: Besides the lack of identification of clouds that originate from convection and those that do not, the other issue I see is the error and uncertainty associated with the trajectory calculation, which increases with TSC. This means that the results for TSC > 5 d may not be meaningful. The error associated with omitting the vertical motions and using the wind averaged between 200 hPa and 300 hPa has been discussed in the manuscript (Section 2.5). However, I am concerned with the error associated with the method of calculating the TSC, in which if a grid box becomes empty, then the value for the TSC in that grid box is taken as the mean value of the surrounding grid boxes (as stated in lines 343–344 in the manuscript). How much is the error that this introduces?*

**Reply**: This is indeed an important point and we have investigated it further to better characterise the related uncertainty. Divergence leaving an empty gridbox occurs on average in 5% of the 0.1 x 0.1 degree gridboxes at each timestep. The mean distribution of the divergence across the tropics is shown in Figure R1 and is now included in the supplementary information.

[Figure]

Figure R1: Frequency of occurrence of divergence producing a gridbox that must be filled by interpolation.

The percentage of gridboxes that diverge is greater in regions of low time since convection (compare R1 to Fig. 5b in the manuscript), where as many as 15% of pixels diverge enough to require interpolation at any timestep. This is unsurprising due to the nature of the diverging wind field at this height and in these regions. However, the nature of this divergence, early in the lifetime of the convective outflow, means that the interpolation used to fill these empty gridboxes is primarily from gridboxes that themselves all have very similar values of TSC. As shown in Fig. R2, in the majority of cases where interpolation is

[Figure]

Figure R2: Histogram of the standard deviation of pixels surrounding a divergent gridbox. These neighbouring pixels are averaged to fill the TSC in the divergent region. The lower quartile is 0.6, the upper quartile is 18.6 and the median standard deviation is 5.08 hours.

required, the neighboring pixels all have the same TSC (a standard deviation of zero). Over half of all cases has a standard deviation of pixels used for the interpolation of 3 or less, such that very little uncertainty is introduced through this interpolation process.

As mentioned above, the actual method for the interpolation was not clear from the description provided. The previous method was used to fill in the divergent gridboxes with the average TSC from all the pixels within the same 1°by 1°gridbox. The updated method uses only an average of the neighbouring 0.1°by 0.1°pixels.

Despite this difference in the interpolation process, there was a neglibile difference in the results presented in this work. Figure R3 shows the difference in the median TSC between the old and new method. The TSC is on average a few hours lower in regions of lower TSC in the new method, as we are only including the neighbouring pixels to the divergent pixels in the interpolation, which usually occurs in regions of low TSC.

The methods section has been rewritten for clarity and to account for the change in method on lines 141-147: *The TSC array is interpolated into missing regions after each advection timestep (Fig. 2). When each 0.1x0.1° pixel in the TSC array is advected forward, some of the trajectories converge to occupy the same pixel, necessarily leaving some pixels without a TSC value. When this*

*occurs, the missing values are interpolated as an average of TSC values around this empty pixel. Between each timestep, divergence leaves approximately 5% of the pixels empty. When two trajectories enter the same pixel, they will proceed to follow the same trajectory ad infinitum due to the deterministic nature of the trajectories. Therefore the trajectory with the smaller TSC value is the one that determines the TSC value at that pixel from that point on. This increases confidence that any high TSC value really represents air at such long timescales since convection.*

[Figure]

Figure R3: Difference between old median TSC and median TSC with new method.
* * *
*: In addition, the number of retrievals decreases with TSC. Figure 7 shows the distribution of clouds along the trajectories, but it has been normalized so we do not see the actual number of retrievals. How many retrievals are there for TSC longer than 5 d? Is this number sufficiently large to guarantee a reliable result? I suppose that a detailed analysis of the error and uncertainty as TSC increases is needed.*

**Reply**: We thank the reviewer for raising this point that the histogram shown in Figure 3 doesn't show the number of retrievals for each TSC value. While the peak in the TSC histogram is close to 0, more than 50% of the TSC values are at times longer than 120hrs. We have added a cumulative distribution to Fig. 3 to make this clearer and noted that there are over 540 million retrievals in the four years TSC data.

Figure R4 shows the labelled histogram that is added to Figure 3 in the paper.
* * *
*: Given the error and uncertainty, should the trajectory analysis be limited for TSC < 5 d? But if so, what would be the new finding of this work compared to Luo and Rossow (2004)'s? Recall that Luo and Rossow (2004) already performed forward trajectory calculations for 5 d starting from deep convective systems to address a similar topic.*

**Reply**: We agree that there is an increased uncertainty at longer TSC values. However we still see value in investigating changes at these longer times. As

[Figure]

Figure R4: Histogram of the TSC distribution for 2007-2010.

demonstrated in Figs. 6 and 8, changes in cloud properties as a function of TSC are still observed at several hundred hours since convection. This alone is a novel result that was not possible with previous methodology.

For the part of this work that focuses on the period within 120 hours (5 days) since convection, there are still significant advances in this work compared to the important work of Luo and Rossow (2004).

Firstly, Luo and Rossow tracked individual convective events for a limited time. This work creates a TSC dataset for the entire tropics such that we do not track individual convection but rather build a composite picture of convective evolution for the entire tropics, enabling the analysis of divergence and intersections of individual trajectories.

This work also introduces novel analysis of the vertical evolution of the cloud field through the use of the DARDAR dataset. This provides a new picture of the evolution of clouds following convection and accounts for the overlying high cloud issue with ISCCP data used in previous work.

Finally, the radiative evolution in this work is novel, isolating the radiative effects of cirrus clouds along the trajectories.

Wording to better highlight the novel aspects of this work is included in lines 83-91: *Most previous work focused on individual mesoscale studies and capped the analysis at shorter time scales in the region of 120 hours. In contrast, this paper presents a method to run a Lagrangian trajectory analysis across the entire tropics at unbounded trajectory lengths at a low computational cost, leading to changes being found in the cloud properties at time scales far beyond 120 hours after the initial convection has dissipated. Used in conjunction with lidar*

*and radar data, the vertical evolution of the cloud properties are characterised as a function of time since convection. The approach in this work builds a composite picture of the lifecycle tropical convective into thin cirrus. This paper also investigates how TSC is a function of latitude, and shows that the changes in the cloud properties along trajectories are a strong function of TSC and are not simply reflective of latitudinal changes as air moves from the tropics to the extratropics. This paper also considers the radiative evolution for just the high clouds along the trajectories.*
* * *
*: In summary, the major issue of this work is the lack of identification of clouds that originate from convection and those that do not. In addition, the error and uncertainty associated with the trajectory calculation method have not been assessed properly. I hope that the authors will consider these points to improve their work before it can be published in ACP.*

**Reply**: We thank the reviewer for these comments. As above, we have now more clearly described the purpose of this study to explain that this work doesn't just look at directly detrained cirrus clouds, and have added new analysis about important uncertainties in the method, along with other changes. We hope that the reviewer is satisfied in our response here and in our improvements to the paper.

**Reviewer 2**
* * *
*: The paper by Horner and Gryspeerdt examines the evolution of anvils cirrus with time using a novel methodology to track air parcels that have been associated with deep convection. They then attempt to discern the evolution of the cloud properties and radiative effects with time. The tendency for the parcels to move poleward from tropical deep convection is an interesting result. The authors find that the longwave heating by cirrus become increasingly important with time as the cirrus associated with deep convection thins. These results are in accordance with previously published results. I feel that this study can be an important contribution to our understanding of the role of cirrus in maintaining the tropical radiative energy budget. However, the authors have failed to adequately explain their methodology and I am left with many questions regarding their results and conclusions. I feel that additional work is needed before this paper is suitable for publication.*

**Reply**: We thank the reviewer for their comments, which we have addressed in turn below.
* * *
*: My main criticism of the paper is that the analysis method is not clearly explained. Their fundamental source of information is ISCCP retrievals. They then combine this with reanalysis, with cloudsat and calipso data, and with CERES flux data. All these data sets have highly variable spatial and temporal resolutions that can influence their results. How they merge this disparate information is not clearly explained. While the use of cloudsat and calipso is*

*innovative, matching the active remote sensor curtains with ISCCP data seems fraught with uncertainty and sampling issues. This is especially true given the 16-day repeat time of cloudsat and calipso. This all needs to be addressed in more detail and sensitivity to assumptions and sampling quantified. A case study would have been interesting to illustrate the methodology.*

**Reply**: We appreciate that the analysis might not have been clearly explained and have modified the wording in section 2.6 on lines 193-218 to improve this. In particular, we have included more details on the datasets used in the methods section and more detail on how we have combined multiple datasets that have different spatial and temporal resolution. In general, we take the nearest TSC and CERES data that matches the time of the ISCCP data. i.e. the TSC and CERES data are sampled every 3 hours to match the ISCCP time. The DARDAR data has a very high temporal and spatial resolution. In order to match this to the 3-hourly, 1 by 1 degree resolution, the DARDAR times are rounded (to the nearest hour) that match the ISCCP time. We have explained this in more detail below.

**Specific comment:**

**Line 54**: *See Schwartz and Mace (2010; doi:10.1029/2009JD012778) who used Cloudsat and Calipso data to examine the mechanisms proposed by Garret and Hartmann et al*

**Reply**: We thank the reviewer for pointing us towards this reference, we have included it in the manuscript on line 55.

**Line 97**: *Reader should not have to dig Teslioudis et al (2021) to understand the data product so more information on the ISCCP H product is needed. Daytime only, I assume? How does it differ from earlier ISCCP products, etc?*

**Reply**: We have included more information, including an updated reference to the ISCCP-H dataset, at lines 104-107: *This work uses the global weather states in Tselioudis et al. (2021), which uses the ISCCP-H 1 by 1° daytime and nighttime 3 hourly dataset. (Rossow et al., 2017) The ISCCP-H differs from prior ISCCP products in numerous ways, most importantly improving the spatial resolution to 1° from 2.5°. A full list of the differences in the data products is given in Rossow et al., (2017). The weather states are identified by clustering the ISCCP tau-CTP joint histogram, resulting in the cluster centroids listed in Table.1.*

**Line 99**: *I don't know what a nearest neighbor algorithm means. More information is needed and a reference would be helpful.*

**Reply**: We have removed 'algorithm' from the description of the nearest neighbour method as it isn't strictly an algorithm we have written. The "nearest neighbour" just means that each 1°by 1°ISCCP gridbox is assigned to the cluster with the properties that most closely represent it. In this work, the centroid value of the cloud regime in 3 dimensions (CTP,albedo,CF) is used to assign the gridbox to the centroid that most closely resembles it in those dimensions. The

text has been reworded at line 107 to make this clearer: *The ISCCP-H dataset is separated into seven distinct cloud regimes by calculating the nearest neighbours of each gridbox and clustering them into the separate regimes depending on the gridbox mean cloud fraction, cloud albedo, and cloud top pressure.*
* * *
**Line 103**: *Given that a realistic optical depth of a convective core on a 1 km scale is on the order of 100, an optical depth of 8.5 is an interesting choice. The old ISCCP data had an optical depth bin of 23. Why is 8.5 chosen here?*

**Reply**: We only want to define the core itself as deep convection, and not any subsequent outflow that has a lower optical thickness. The ISCCP threshold of 23 applies at the retrieval scale (where as the reviewer states, a deep convective cloud might have a very large optical depth). In this work, we are identifying deep convection based on 1°by 1°data, such that the average optical depth will be smaller - the deep convection cluster centroid has an optical depth of only 10.5 (Tab. 1).

Assigning gridboxes to clusters doesn't have an inbuilt threshold for optical depth or CTP. We found that without setting this condition we risked missing the evolution of the some of the anvil as it was categorised as being 'deep convection'. This is now made clear in lines 118-121: *Additional conditions are applied to isolate the convective cores, requiring a $\tau_c > 8.5$ (albedo > 0.5) and a cloud top temperature (CTT) < 220K. Only the very brightest, thickest cores of the convective clouds are categorised as DCC. If these conditions aren't imposed then thick anvil cirrus are included as part of the convective cores, and the ability to investigate their temporal development is reduced.*
* * *
**Line 105**: *Where does this expression come from? What is 0.895?*

**Reply**: This expression comes from the ISCCP simulator from Jakob and Klein and accounts for scale effects in the non-linear relationship between albedo and cloud optical depth. A reference has been included on line 111: *...as defined in the ISCCP simulator (Klein and Jakob, 1999)*
* * *
**Line 105**: *Assuming a CloudSat footprint of 2km, and a 100x100 km ISCCP footprint, there are 2500 2x2km pixels in an isccp grid box. A diagonal across the middle of the box (the maximum that could be achieved) gives about 450 pixels. How is this disparity in sampling accounted for in a highly variable field? Are the data averaged? Since cloudsat makes 14 passes over the tropics in a day, how is the disparity in temporal sampling accounted for? The authors present only 2 years of data.*

**Reply**: The DARDAR data is not pre-averaged across the 1 by 1° gridbox. This means that there are many CloudSat overpasses corresponding to the same TSC value, and all these CloudSat pixels contribute to the average CloudSat profile for each TSC bin.

The wording of the text has been modified to make this clearer on lines 202-206: *The vertical evolution of the cirrus is investigated using the DARDAR dataset, a combination of the CloudSat radar and CALIPSO lidar. The overpass locations of DARDAR are matched within the hour to the TSC at that location, at 1 by 1 degree resolution. A 1°by 1°gridbox can contain many DARDAR*

*retrievals, each of these DARDAR retrievals is assigned the same TSC value - that of the 1°by 1°gridbox. These DARDAR retrievals then all contribute equally to the analysis in the relevant TSC bin.*
* * *
**Line 193**: *What is the resolution of the CERES data? The native footprint of ceres is 20 km. Is the CERES data averaged? Also, CERES passes over the tropics 14 times per day on MODIS. Does it provide global coverage in a day? ISCCP H has 3 hourly resolution. How are these data products merged?*

**Reply**: The CERES SYN-1deg dataset is at a 1 by 1° resolution. CERES data provides global coverage at 1 hourly temporal resolution by combining MODIS satellite data with other geostationary satellites. The data products are merged by sampling the CERES data every 3 hours to coincide with the ISCCP data time. The wording of the Data section (lines 94-101) is improved to make this clearer: *Observational data from the 3-hourly International Satellite Cloud Climatology Project (ISCCP) H dataset at 1x1° is used to define locations of deep convection (Rossow et al., 2017). ECMWF ERA5 reanalysis wind fields are used in the trajectory analysis (Hersbach et al., 2018) and to characterise cloud properties. To examine the evolution of the radiative properties, the CERES SYN1deg L3 LW and SW TOA fluxes are used (NASA/LARC/SD/ASDC, 2017). The CERES SYN1deg product combines MODIS and geostationary satellite data to provide global coverage at a 1x1° resolution and 1 hourly temporal resolution. The vertical evolution of the cloud is investigated by utilising the DARDAR dataset, which is an ice cloud retrieval product that combines measurements from the CloudSat radar and CALIPSO lidar (Delanoë and Hogan, 2008; Sourdeval et al., 2018). The period of study in this paper is 2008-2010 inclusive..*
* * *
**Line 224**: *If the mean TSC is 180 hours, it would seem that all air has been in contact with convection by day 5. At some point it seems impossible to associate the clouds with convection.*

**Reply**: We are not entirely sure what the reviewer means by this. The reviewer makes a valid point that it becomes more difficult to associate the clouds with convection at longer times since convection, and we have been conscious of not attributing changes in cloud properties at longer TSC to the properties of the initial convective event itself. Nonetheless, TSC provides an important metric to investigate how cloud properties change as a function of time.
* * *
**Line 227**: *The authors claim that the TSC is relevant to 360 hours. I don't seem how this can be surmised. the air moves more and more into regions of subsidence with time. large-scale subsidence may be as much responsible for the change as some sort of post convection decay of anvil cirrus. Can this possibility be discounted? I think the tendency for the cirrus move out of the tropics with time into the large-scale subsidence as illustrated in Figure 5 is one of the more interesting results presented in the paper.*

**Reply**: The reviewer is correct that large scale subsidence may be responsible for the change rather than convective decay, particularly at long timescales. We were trying to to show how TSC can be used as a metric to characterise changes in cloud properties along a trajectory, not that the properties are changing due

[Figure]

Figure R5: Zonally averaged DARDAR cloud fraction for 12 hour TSC bins.

to the initial convection itself. The word "relevant" has been removed and L243 has been reworded to emphasise that TSC is merely a useful metric for changes in cloud properties. In addition, section 3.6 has been added to the analysis that includes the zonally averaged DARDAR cloud amount, isolating the dependence of cloud amount on TSC from latitude. Future work will assess the impact of the initial convection on the evolution of high clouds as a function of TSC.

We note that latitudinal variations are not purely responsible for the observed changes in cloud properties. Figure R5 shows the zonally averaged vertical cloud amount for increasing TSC bins. It is clear that at very low TSC values, the vertical cloud fraction is almost only a function of TSC, with very little variation in cloudiness profiles as a function of latitude (Fig. R1a). The cloud amount is consistent across all latitudes for a given TSC at early TSC values, showing that TSC is the controlling factor in the vertical evolution of clouds along TSC trajectories, and not just latitude. Latitude becomes important at longer TSC values, where the TSC becomes less central a controlling factor for the vertical structure of high clouds. This plot has been included in the original paper.
* * *
**Line 265**: *The authors need to explain how the DARDAR product is able to deduce Ni when the anvils are optically thick for up to 1-day TSC? This implies that only the radar is able to provide credible information to the retrieval. Even when optically thin, the authors need to comment on the validity and uncertainty of the Ni retrieval given that that the lidar is a cross sectional area constraint and the radar a mass-squared constraint. With the shape of the ice crystals unknown, converting the remote sensing into Ni is fraught with uncertainty. Might it be possible that any Ni signal the authors infer are due to changes in ice crystal habit over time? An evolution in habit as the anvils age would result in a time-dependent ratio of mass and area that is not accounted*

*for in DARDAR.*

**Reply**: Producing a new validation of the DARDAR product is out of scope for this work, but we note that for clouds colder than about -40C, the DARADR Ni retrieval compares very favourably to in-situ aircraft measurements (Sourdeval et al., 2018) While the accuracy is reduced in warmer clouds, these are not the primary focus of the study.

However, as the reviewer notes, there are still considerable uncertainties surrounding the retrieval, particularly with particle shape. Although this is approximately a function of temperature, variations in shape over time could produce an apparent change in Ni. While this is planned to be investigated in future work, we have added a note to explain these caveats in the discussion at lines 380-384: *While there are significant changes in DARDAR ice properties as a function of TSC (Fig. 8), there are still considerable uncertainties surrounding the retrieval, particularly at warmer temperatures. The DARDAR-Nice Ni retrieval compares favourably to in-situ measurements at the cirrus temperatures that are the focus of this work (Sourdeval et al., 2018). However, the potential temporal variation of factors such as particle shape might introduce TSC-dependent biases in the Ni retrieval, producing an apparent change in Ni. This is will be investigated in future work through a comparison with aircraft data..*

**Bibliography**

Delanoë, J. and Hogan, R. J.: A variational scheme for retrieving ice cloud properties from combined radar, lidar, and infrared radiometer, Journal of Geophysical Research: Atmospheres, 113, https://doi.org/10.1029/2007JD009000, 2008.

Hersbach, H., Bell, B., Berrisford, P., Biavati, G., Horányi, A., Muñoz Sabater, J., Nicolas, J., Peubey, C., Radu, R., Rozum, I., Schepers, D., Simmons, A., Soci, C., Dee, D., and Thépaut, J.-N.: ERA5 hourly data on pressure levels from 1959 to present., https://doi.org/10.24381/cds.bd0915c6, 2018.

Klein, S. A. and Jakob, C.: Validation and Sensitivities of Frontal Clouds Simulated by the ECMWF Model, Monthly Weather Review, 127, 2514 – 2531, https://doi.org/https://doi.org/10.1175/1520-0493(1999)127⟨2514: VASOFC⟩2.0.CO;2, 1999.

NASA/LARC/SD/ASDC: CERES and GEO-Enhanced TOA, Within-Atmosphere and Surface Fluxes, Clouds and Aerosols 1-Hourly Terra-Aqua Edition4A, URL `10.5067/TERRA+AQUA/CERES/SYN1DEG-1HOUR_L3.004A`, 2017.

Rossow, W., Golea, V., Walker, A., Knapp, K., Young, A., Hankins, B., and Inamdar, A.: International Satellite Cloud Climatology Project (ISCCP) Climate Data Record, H-Series, https://doi.org/10.7289/V5QZ281S, 2017.

Sourdeval, O., Gryspeerdt, E., Krämer, M., Goren, T., Delanoë, J., Afchine, A., Hemmer, F., and Quaas, J.: Ice crystal number concentration estimates from lidar–radar satellite remote sensing – Part 1: Method and evaluation, Atmospheric Chemistry and Physics, 18, 14 327–14 350, https://doi.org/10.5194/acp-18-14327-2018, 2018.

Tselioudis, G., Rossow, W. B., Jakob, C., Remillard, J., Tropf, D., and Zhang, Y.: Evaluation of Clouds, Radiation, and Precipitation in CMIP6 Models Using Global Weather States Derived from ISCCP-H Cloud Property Data, Journal of Climate, 34, 7311 – 7324, https://doi.org/10.1175/JCLI-D-21-0076.1, 2021.

---

## Referee Report (RR1)

Second review of "The evolution of deep convective systems and their associated cirrus outflows"
by George Horner and Edward Gryspeerdt
* * *
I appreciate the authors' additional anylysis to differentiate between detrained cirrus and in-situ cirrus that form after the detrained cirrus dissipated. However, I am concerned that the method used to do this is not correct. The authors stated that whenever the cloud fraction of the detrained cirrus has dropped below 10 % for the first time, then any subsequent cirrus are considered to form in situ. The authors stated that this method is similar to that used by Luo and Rossow (2004), but I don't think that this is the case.

For the purpose of this discussion, I have taken the liberty to copy the text describing the method to differentiate detrained and in-situ cirrus by Luo and Rossow (2004) here: "We consider two types of tropical cirrus based on their relationship to convective systems: detrainment cirrus and in situ cirrus. The life cycle for detrainment cirrus is defined by a monotonic decrease of cirrus amount along a forward trajectory like this: last convection becoming some cirrus becoming zero cirrus. The length of this sequence defines the detrainment cirrus lifetime. For in situ cirrus, the sequence is appearance of cirrus with first an increase and then a decrease of cirrus amount like this: zero cirrus (observed at least 1 day after convection) becoming some cirrus becoming zero cirrus. The requirement that the air parcel encounter clear skies after convection ensures the separate evolution of the in situ cirrus." ... "In practice, we take zero cirrus to be 1/5 of the maximum cirrus cloud amount along the lifecycle sequence. For example, when cirrus overcast decays below 0.20, we call this the end of the life cycle. But if the maximum cirrus amount along a selected trajectory is only 0.20, then it has to drop below 0.04 to end the life cycle."

Figures 8(a) and 8(b) of this manuscript show that the cloud fraction at TSC = 0 is 0.3 on average. Therefore, if we apply the method used by Luo and Rossow (2004), then on average the detrained cirrus would disappear when the cloud fraction is reduced below $0.3/5 = 0.06 = 6\%$, not 10 %. Moreover, the maximum cloud fraction obtained at TSC = 0 may be different for different trajectories, so the threshold to determine the disappearance of the detrained cirrus would change from one trajectory to the next.

Figure 3(b) does not make sense to me, either. For TSC less than about 10–15 hours, the number of counts of in-situ cirrus is not visible because it is shown beneath the number of counts of detrained cirrus, but let me assume that in this figure these two numbers are equal for TSC less than about 10–15 hours. If this is what's plotted in the figure, I don't understand why there would be so many in-situ cirrus already for small TSC.

The authors aim to demonstrate that convection has a significant long-lasting impact on the properties of clouds. However, in Fig. 7(b), the clouds at 370 hPa do not appear to be connected to the initial convection. Please see also my comment on the original version of the manuscript about these clouds. It appears to me from Fig. 7(b) that these clouds are neither convectively detrained nor formed in-situ from the moisture perturbation brought about by the convection. The presence of these clouds certainly affects the average cloud properties, for example, cloud radiative effect (CRE). It follows that the CRE shown in Fig. 10 is not purely from clouds associated with convection and cannot be used to demonstrate the impact of convection on CRE. I believe this is a major issue with this study.

Finally, a specific issue to be fixed is the caption of Fig. 9. In the current version of the manuscript, it is incorrectly identical to the caption of Fig. 10.

**References**

Z. Luo and W. B. Rossow. Characterizing Tropical Cirrus Life Cycle, Evolution, and Interaction with Upper-Tropospheric Water Vapor Using Lagrangian Trajectory Analysis of Satellite Observations. *J. Climate*, 17(23):4541–4563, 2004.

---

## Author Response (AR2)

**Editors Comments**
* * *
*: Dear Dr. Horner, Thank you for your response to reviewers file. Please revise it so that it explicitly includes, wherever a substantial modification is made, the actual change that is made in the body of the document. This facilitates the review process immensely. Further, I am concerned by the response to Reviewer 1 in particular, who requested separate treatment of clouds that are not explicitly associated with convection, presumably referring to those that might be associated with large scale synoptic uplift or gravity waves. Also, the response to this comment is seemingly contradictory: "This study instead investigates the evolution of air parcels (and the associated cloud properties) advected from convective cores. While some of the clouds studied will have detrained from convection, as the reviewers note, this study is not limited to these clouds.". Hopefully these matters can be more fully addressed. Regards, Tim Garrett*

**Reply**: We thank the editor for raising these points. We have made sure it is clear in our reviewer response where we have made substaintial modifications to the paper by including these modifications in **bold** in our reply, with reference to the specific line numbers that these changes occur in the paper itself.

To address the reviewers question about the relative importance of detrained vs in situ cirrus, we conducted an additional analysis, similar to that in Luo and Rossow (2004). This allows us to identify detrained cirrus, and gridboxes that might potentially contain in situ-formed cirrus. We note that the properties of both detrained and in situ cirrus change as a function of TSC, although the detrained cirrus are concentrated at lower TSCs (as expected). We thanks the reviewer (and editor) for suggesting the analysis, as it is a very useful addition to the paper which helps in interpreting the results shown. We hope this response is satisfactory for the reviewers.

**Reviewer 1**
* * *
*: The authors aim to study the evolution of deep convective systems and their associated cirrus outflows, as the title of the manuscript suggests. As far as I understand, the most significant finding of this work compared to existing literature is the sustained warming effect of convectively detrained cirrus clouds. The authors argue that this effect is sustained past 5 d since the time of the last deep convective event. My main concern is whether the cirrus clouds captured along the trajectories, especially at an increasingly long time since convection (TSC), are actually associated with convection. If the goal of the paper is to evaluate the radiative effect of the clouds associated with convection, the authors would need to properly separate the clouds associated with convection and those that are not. The authors have not provided the analysis to do this.*

**Reply**: We thank the reviewer for their comments which we address in turn

below. The goal of the paper is not to track the evolution of exclusively detrained clouds from convection, as this has been done before in mesoscale studies (e.g. Luo and Rossow, 2004).

This study instead investigates the evolution of air parcels (and the associated cloud properties) advected from convective cores. While some of the clouds studied will have detrained from convection, as the reviewers note, this study is not limited to these clouds. To make this distinction more explicit, we have included a section where we look at the proportion of cirrus clouds along trajectories that are directly detrained from convection, verses those which form in situ along trajectories. We can now say for each TSC value whether a cirrus cloud is detrained from convection verses formed in in situ.

We have included a new section, 2.4, where we introduce a flag for identifying regions where the cirrus would be in situ in origin verses detrained on lines 156-164: ***Not all of the cirrus along the advected air parcels will have come directly from the convective cores. Therefore it is necessary to differentiate between detrained cirrus verses in situ cirrus that formed at any point after the detrained cirrus dissipated. To do this a 'cirrus type' value is assigned to each gridbox where detrained cirrus is present. This is defined as there being a high cloud fraction (using the ISCCP histogram) greater than 10% at any point after TSC-0. Once the cloud fraction drops below 10%, the cirrus type for that grid box becomes in situ. This means that any cirrus cloud that then forms along this advected air parcel after the detrained cirrus has dissipated is flagged as an in-situ cirrus. This cirrus type value doesn't necessitate that a cirrus cloud be present in the gridbox, just that the cloud fraction of the detrained cirrus has dropped below 10% for the first time, meaning that any subsequent cirrus that forms is in situ. This is a similar method used to Luo and Rossow (2004).***

In addition, we have updated Figure 3b) (Figure R4 in the review) to show what percentage of the TSC gridboxes would contain detrained vs in situ cirrus. We can see that most of the detrained cirrus is confined to the first 12-24 hours from convection (as expected), and the proportion of in situ origin cirrus increases after this point.

In contrast to previous studies, this work uses time since convection (TSC) to characterise all tropical high clouds and their radiative properties, irrespective of whether they were directly detrained from convective cores. We also include new analysis of how TSC varies as a function of latitude. We have modified the text in the introduction, methods, discussion and conclusion to better highlight this difference to previous work.

During the preparation of this response, the method for interpolating between diverging trajectories has been modified to be more consistent with the description in the methods. These changes are discussed in detail on Page 3 of the response, but have a minimal impact on the results presented in this work.

We have also harmonised some of the terminology, using 'pixel' to refer to an element of the 0.1° by 0.1° used for the TSC calculation, where 'gridbox' is

an element of the 1° by 1° grid.

**General comments:**

*: The main argument that the authors gave to justify that deep convection leads to the formation of the clouds along the trajectories is the expectation of changes in cloud properties with the TSC. Specifically, as the anvil cirrus decays into thin cirrus, it is expected that the cloud fraction decreases, the cloud top pressure increases (as clouds descend to lower altitudes), and the cloud optical depth decreases. These behaviours are seen clearly in the data the first 24 h (Fig. 7), but then the signals become weak (Fig. 6) and/or disconnected in time and/or space (Fig. 7).*

**Reply**: We agree that the signal becomes weaker at longer times since convection, but suggest that this is expected. As the reviewer notes, the behaviour of clouds at longer times is less likely to be related to the properties of the initial convection, and we have now shown this in our separate analysis of in situ vs detrained cirrus in section 2.4 and with the updated Figure 3b). However, we note that cloud properties still vary as a function of TSC even at very large TSC values, where the cirrus has formed in situ along trajectories. We believe it is still interesting and worthy of investigation that cloud properties change as a function of TSC, as we might expect that cirrus formed in situ along trajectories from convection may still be influenced by the air masses advected from the convection even if they have not directly detrained out of the convective core.

**We have also included a new section of analysis and extra plot into the paper that investigates the role that latitude plays in the evolution of the cloud properties, and how the influence of TSC changes as air parcels get further from convection (i.e. at large TSC values).** These results are particularly important to isolate the direct influence of TSC on the changes in cloud properties along trajectories, and highlights that these changes aren't merely due to latitudinal influence on cloud properties. **This begins on line 333 in the section 'TSC as a function of latitude' :** *"It is clear from Fig. 5b) that on average TSC increases with latitude, expected from the large-scale behaviour of the Hadley circulation..."*

*: In Fig. 7(b), high clouds are seen at 250 hPa, 310 hPa, and 370 hPa, but only the clouds at 250 hPa and 310 hPa appear to be connected to the convective anvil, while the clouds at 370 hPa do not. Where do the clouds at 370 hPa come from? It is odd that clouds appear at 370 hPa at TSC = 10 h before the thick convective anvil above dissipates. Is this an artifact of the data and/or the method? If so, is the data and/or method trustworthy? In addition, the cloud top pressures of the high clouds at respectively 250 hPa, 310 hPa, and 370 hPa do not increase with time (Fig. 7b) as expected, and the optical depth of the clouds does not decrease with time either after about 30 h (Figs. 7c and 7d).*

**Reply**: It is difficult to make a definite statement about the original of the

clouds at 370hPa due to the overlying high cloud - but we note that they become visible as the thick initial layer of cloud at 200-250hPa dissipates. This interpretation is supported by the DARDAR cross section plots (Figs. 8a,b), which do not show the appearance of a new layer at 8/9km (approximately 370hPa). We have modified the text to make it clearer that the these clouds may not be directly detrained to the initial convection on lines 285-288: *There is a significant cloud layer at 370hPa that becomes visible as the high cloud dissipates. These are likely clouds not directly detrained from convection that become visible as the cloud fraction of the anvil cloud above decreases, and any changes in the properties of these clouds, particularly at large TSC values, are not suggested to be directly related to the initial convection.*
* * *
*: Figures 8(d) and 8(f) show that the convective outflow decays within 36 h–42 h (line 285 in the manuscript). Subsequent anomalies in ice number concentration are not discussed in the manuscript, but they appear to me to be disconnected to the convective outflow anvil in these figures. In short, both Figs. 7 and 8 suggest that a large number clouds along the trajectories may not be associated with convection (or at least not directly), particularly for long TSC.*

**Reply**: Our new analysis of the in situ v detrained cirrus clouds shows that a number of clouds along the trajectories are indeed not directly detrained from the initial convective event. The number of in situ cirrus clouds along trajectories increases with TSC as the detrained cirrus dissipate. However it is still of scientific value to investigate how cloud properties from clouds not directly detrained from deep convection may change as a function of time since convection, due to the impact of convection on the properties of the UTLS region, including humidity and aerosol environment. As highlighted above, this point has been clarified on lines 285-288 with a discussion section included on line 329 and a plot included that shows how the cloud distribution is independent of latitude for a given TSC value. **We have also included our new analysis in section 2.4 and an updated Figure 3. that explicitly identifies in situ verses detrained cirrus clouds.**
* * *
*: Besides the lack of identification of clouds that originate from convection and those that do not, the other issue I see is the error and uncertainty associated with the trajectory calculation, which increases with TSC. This means that the results for TSC > 5 d may not be meaningful. The error associated with omitting the vertical motions and using the wind averaged between 200 hPa and 300 hPa has been discussed in the manuscript (Section 2.5). However, I am concerned with the error associated with the method of calculating the TSC, in which if a grid box becomes empty, then the value for the TSC in that grid box is taken as the mean value of the surrounding grid boxes (as stated in lines 343–344 in the manuscript). How much is the error that this introduces?*
**Reply**: This is indeed an important point and we have investigated it further to better characterise the related uncertainty. Divergence leaving an empty gridbox occurs on average in 5% of the 0.1 x 0.1 degree gridboxes at each timestep. The

mean distribution of the divergence across the tropics is shown in Figure R1 and is now included in the supplementary information.

[Figure]

Frequency of gridboxes containing nan pixels / %

Figure R1: Frequency of occurrence of divergence producing a gridbox that must be filled by interpolation.

[Figure]

Figure R2: Histogram of the standard deviation of pixels surrounding a divergent gridbox. These neighbouring pixels are averaged to fill the TSC in the divergent region. The lower quartile is 0.6, the upper quartile is 18.6 and the median standard deviation is 5.08 hours.

The percentage of gridboxes that diverge is greater in regions of low time since convection (compare R1 to Fig. 5b in the manuscript), where as many as 15% of pixels diverge enough to require interpolation at any timestep. This

is unsurprising due to the nature of the diverging wind field at this height and in these regions. However, the nature of this divergence, early in the lifetime of the convective outflow, means that the interpolation used to fill these empty gridboxes is primarily from gridboxes that themselves all have very similar values of TSC. As shown in Fig. R2, in the majority of cases where interpolation is required, the neighboring pixels all have the same TSC (a standard deviation of zero). Over half of all cases has a standard deviation of pixels used for the interpolation of 3 or less, such that very little uncertainty is introduced through this interpolation process.

As mentioned above, the actual method for the interpolation was not clear from the description provided. The previous method was used to fill in the divergent gridboxes with the average TSC from all the pixels within the same 1°by 1°gridbox. The updated method uses only an average of the neighbouring 0.1°by 0.1°pixels.

Despite this difference in the interpolation process, there was a neglibile difference in the results presented in this work. Figure R3 shows the difference in the median TSC between the old and new method. The TSC is on average a few hours lower in regions of lower TSC in the new method, as we are only including the neighbouring pixels to the divergent pixels in the interpolation, which usually occurs in regions of low TSC.

The methods section has been rewritten for clarity and to account for the change in method on lines 141-147: ***The TSC array is interpolated into missing regions after each advection timestep (Fig. 2). When each 0.1x0.1° pixel in the TSC array is advected forward, some of the trajectories converge to occupy the same pixel, necessarily leaving some pixels without a TSC value. When this occurs, the missing values are interpolated as an average of TSC values around this empty pixel. Between each timestep, divergence leaves approximately 5% of the pixels empty. When two trajectories enter the same pixel, they will proceed to follow the same trajectory ad infinitum due to the deterministic nature of the trajectories. Therefore the trajectory with the smaller TSC value is the one that determines the TSC value at that pixel from that point on. This increases confidence that any high TSC value really represents air at such long timescales since convection.***
* * *
*: In addition, the number of retrievals decreases with TSC. Figure 7 shows the distribution of clouds along the trajectories, but it has been normalized so we do not see the actual number of retrievals. How many retrievals are there for TSC longer than 5 d? Is this number sufficiently large to guarantee a reliable result? I suppose that a detailed analysis of the error and uncertainty as TSC increases is needed.*

**Reply**: We thank the reviewer for raising this point that the histogram shown in Figure 3 doesn't show the number of retrievals for each TSC value. While the peak in the TSC histogram is close to 0, more than 50% of the TSC values are at times longer than 120hrs. We have added a cumulative distribution to

[Figure]

Figure R3: Difference between old median TSC and median TSC with new method.

Fig. 3 to make this clearer and noted that there are over 540 million retrievals in the four years TSC data.

**Figure R4 shows the labelled histogram that is added to Figure 3 in the paper showing the new analysis that identifies the gridboxes potentially containing detrained cirrus clouds verses in situ cirrus clouds.**

[Figure]

Figure R4: Histogram of the TSC distribution for 2007-2010 which highlights the fraction of gridboxes that would contain a detrained cirrus cloud verses the in situ cirrus cloud.

*: Given the error and uncertainty, should the trajectory analysis be limited for TSC < 5 d? But if so, what would be the new finding of this work com-*

*pared to Luo and Rossow (2004)'s? Recall that Luo and Rossow (2004) already performed forward trajectory calculations for 5 d starting from deep convective systems to address a similar topic.*

**Reply**: We agree that there is an increased uncertainty at longer TSC values. However, the changes at longer timescales are critical for the climate, given the large fraction of points at a TSC¿120 hours. As demonstrated in Figs. 6 and 8, changes in cloud properties as a function of TSC are still observed at several hundred hours since convection. This alone is a novel result that was not possible with previous methodology. We also note that this is not purely a function of latitude - **see the new work in section 3.5 and Figure 9.**

For the part of this work that focuses on the period within 120 hours (5 days) since convection, there are still significant advances in this work compared to the important work of Luo and Rossow (2004).

Firstly, Luo and Rossow tracked individual convective events for a limited time. This work creates a TSC dataset for the entire tropics such that we do not track individual convection but rather build a composite picture of convective evolution for the entire tropics, enabling the analysis of divergence and intersections of individual trajectories.

This work also introduces novel analysis of the vertical evolution of the cloud field through the use of the DARDAR dataset. This provides a new picture of the evolution of clouds following convection and accounts for the overlying high cloud issue with ISCCP data used in previous work.

Finally, the radiative evolution in this work is novel, isolating the radiative effects of cirrus clouds along the trajectories.

Wording to better highlight the novel aspects of this work is included in lines 83-91: ***Most previous work focused on individual mesoscale studies and capped the analysis at shorter time scales in the region of 120 hours. In contrast, this paper presents a method to run a Lagrangian trajectory analysis across the entire tropics at unbounded trajectory lengths at a low computational cost, leading to changes being found in the cloud properties at time scales far beyond 120 hours after the initial convection has dissipated. Used in conjunction with lidar and radar data, the vertical evolution of the cloud properties are characterised as a function of time since convection. The approach in this work builds a composite picture of the lifecycle tropical convective into thin cirrus. This paper also investigates how TSC is a function of latitude, and shows that the changes in the cloud properties along trajectories are a strong function of TSC and are not simply reflective of latitudinal changes as air moves from the tropics to the extratropics. This paper also considers the radiative evolution for just the high clouds along the trajectories.***

*: In summary, the major issue of this work is the lack of identification of clouds that originate from convection and those that do not. In addition, the error and uncertainty associated with the trajectory calculation method have not been assessed properly. I hope that the authors will consider these points to improve*

*their work before it can be published in ACP.*

**Reply**: We thank the reviewer for these comments which have greatly improved our work. As above, we have now more clearly described the purpose of this study to explain that this work doesn't just look at directly detrained cirrus clouds, and have added a section that shows that we are aware of the distinction between the detrained cirrus and in situ cirrus that form along trajectories in section 2.4 and Figure 3b). We have added new analysis about important uncertainties in the method, along with other changes. We hope that the reviewer is satisfied in our response here and in our improvements to the paper.

**Reviewer 2**
* * *
*: The paper by Horner and Gryspeerdt examines the evolution of anvils cirrus with time using a novel methodology to track air parcels that have been associated with deep convection. They then attempt to discern the evolution of the cloud properties and radiative effects with time. The tendency for the parcels to move poleward from tropical deep convection is an interesting result. The authors find that the longwave heating by cirrus become increasingly important with time as the cirrus associated with deep convection thins. These results are in accordance with previously published results. I feel that this study can be an important contribution to our understanding of the role of cirrus in maintaining the tropical radiative energy budget. However, the authors have failed to adequately explain their methodology and I am left with many questions regarding their results and conclusions. I feel that additional work is needed before this paper is suitable for publication.*

**Reply**: We thank the reviewer for their comments, which we have addressed in turn below.
* * *
*: My main criticism of the paper is that the analysis method is not clearly explained. Their fundamental source of information is ISCCP retrievals. They then combine this with reanalysis, with cloudsat and calipso data, and with CERES flux data. All these data sets have highly variable spatial and temporal resolutions that can influence their results. How they merge this disparate information is not clearly explained. While the use of cloudsat and calipso is innovative, matching the active remote sensor curtains with ISCCP data seems fraught with uncertainty and sampling issues. This is especially true given the 16-day repeat time of cloudsat and calipso. This all needs to be addressed in more detail and sensitivity to assumptions and sampling quantified. A case study would have been interesting to illustrate the methodology.*

**Reply**: We appreciate that the analysis might not have been clearly explained and have modified the wording in **section 2.7 on lines 215-229** to improve this. In particular, we have included more details on the datasets used in the methods section and more detail on how we have combined multiple datasets that have different spatial and temporal resolution. In general, we take the nearest TSC and CERES data that matches the time of the ISCCP data. i.e.

the TSC and CERES data are sampled every 3 hours to match the ISCCP time. The DARDAR data has a very high temporal and spatial resolution. In order to match this to the 3-hourly, 1 by 1 degree resolution, the DARDAR times are rounded (to the nearest hour) that match the ISCCP time.: ***The vertical profile of the cirrus is investigated using the DARDAR dataset, a combination of the CloudSat radar and CALIPSO lidar (Delanoë and Hogan, 2008).. The overpass locations of DARDAR are matched within the hour to the TSC at that location, at 1x1° resolution. A 1x1° can contain many DARDAR retrievals, each of these DARDAR retrievals is assigned the same TSC value - that of the 1x1° gridbox. These DARDAR retrievals then all contribute equally to the analysis in the relevant TSC bin The DARDAR cloud mask is used to filter out aerosol, ground, and unknown retrievals. The DARDAR-Nice product (Sourdeval et al., 2018) is used to provide the vertical profile of ice crystal number concentration ($N_i$) for crystals larger than 5um ($N_i^5$) and larger than 100um ($N_i^{100}$). The radiative evolution of convection, from thick convective core to thin anvil cirrus, is investigated by analysing the CERES SYN1deg L3 LW and SW TOA fluxes (NASA/LARC/SD/ASDC, 2017). The CRE is calculated as the all sky minus clear sky radiative fluxes from the CERES data. In order to only look at the radiative evolution of the highest clouds, regions where the cloud fraction of the low clouds (as defined by the ISCCP joint histograms) is less than 1% are classified as "high cloud only "gridboxes. For the cloudiest regions in the tropics (which are also the regions of the lowest TSC), there is often no hourly outgoing clear sky LW data. For these gridboxes, the 3 monthly seasonal average outgoing clear sky LW for these gridboxes is used, in order to calculate a value for the CRE. The ISCCP histograms used to isolate the high clouds are only available during daylight, such that the high cloud only CRE is only available during daylight. However, the SW is known to be zero during the night, so the nighttime values for the SW CRE are included in the analysis as zero. The daytime LW values are assumed representative of the nighttime values.***

**Specific comment:**
* * *
**Line 54**: *See Schwartz and Mace (2010; doi:10.1029/2009JD012778) who used Cloudsat and Calipso data to examine the mechanisms proposed by Garret and Hartmann et al*
**Reply**: We thank the reviewer for pointing us towards this reference, we have included it in the manuscript on line 55.
* * *
**Line 97**: *Reader should not have to dig Teslioudis et al (2021) to understand the data product so more information on the ISCCP H product is needed. Daytime only, I assume? How does it differ from earlier ISCCP products, etc?*

**Reply**: We have included more information, including an updated reference to the ISCCP-H dataset, at lines 104-107: ***This work uses the global weather states in Tselioudis et al. (2021), which uses the ISCCP-H 1 by 1° daytime and nighttime 3 hourly dataset. (Rossow et al., 2017) The ISCCP-H differs from prior ISCCP products in numerous ways, most importantly improving the spatial resolution to 1° from 2.5°. A full list of the differences in the data products is given in Rossow et al., (2017). The weather states are identified by clustering the ISCCP tau-CTP joint histogram, resulting in the cluster centroids listed in Table.1.***
* * *
**Line 99**: *I don't know what a nearest neighbor algorithm means. More information is needed and a reference would be helpful.*

**Reply**: We have removed 'algorithm' from the description of the nearest neighbour method as it isn't strictly an algorithm we have written. The "nearest neighbour" just means that each 1°by 1°ISCCP gridbox is assigned to the cluster with the properties that most closely represent it. In this work, the centroid value of the cloud regime in 3 dimensions (CTP,albedo,CF) is used to assign the gridbox to the centroid that most closely resembles it in those dimensions. The text has been reworded at line 107 to make this clearer: ***The ISCCP-H dataset is separated into seven distinct cloud regimes by calculating the nearest neighbours of each gridbox and clustering them into the separate regimes depending on the gridbox mean cloud fraction, cloud albedo, and cloud top pressure.***
* * *
**Line 103**: *Given that a realistic optical depth of a convective core on a 1 km scale is on the order of 100, an optical depth of 8.5 is an interesting choice. The old ISCCP data had an optical depth bin of 23. Why is 8.5 chosen here?*

**Reply**: We only want to define the core itself as deep convection, and not any subsequent outflow that has a lower optical thickness. The ISCCP threshold of 23 applies at the retrieval scale (where as the reviewer states, a deep convective cloud might have a very large optical depth). In this work, we are identifying deep convection based on 1°by 1°data, such that the average optical depth will be smaller - the deep convection cluster centroid has an optical depth of only 10.5 (Tab. 1).

Assigning gridboxes to clusters doesn't have an inbuilt threshold for optical depth or CTP. We found that without setting this condition we risked missing the evolution of the some of the anvil as it was categorised as being 'deep convection'. This is now made clear in lines 118-121: ***Additional conditions are applied to isolate the convective cores, requiring a $\tau_c > 8.5$ (albedo $> 0.5$) and a cloud top temperature (CTT) $< 220K$. Only the very brightest, thickest cores of the convective clouds are categorised as DCC. If these conditions aren't imposed then thick anvil cirrus are included as part of the convective cores, and the ability to investigate their temporal development is reduced.***
* * *
**Line 105**: *Where does this expression come from? What is 0.895?*

**Reply**: This expression comes from the ISCCP simulator from Jakob and Klein and accounts for scale effects in the non-linear relationship between albedo and cloud optical depth. A reference has been included on line 111: ***...as defined in the ISCCP simulator (Klein and Jakob, 1999)***
* * *
***Line 105****: Assuming a CloudSat footprint of 2km, and a 100x100 km ISCCP footprint, there are 2500 2x2km pixels in an isccp grid box. A diagonal across the middle of the box (the maximum that could be achieved) gives about 450 pixels. How is this disparity in sampling accounted for in a highly variable field? Are the data averaged? Since cloudsat makes 14 passes over the tropics in a day, how is the disparity in temporal sampling accounted for? The authors present only 2 years of data.*

**Reply**: The DARDAR data is not pre-averaged across the 1 by 1° gridbox. This means that there are many CloudSat overpasses corresponding to the same TSC value, and all these CloudSat pixels contribute to the average CloudSat profile for each TSC bin.

The wording of the text has been modified to make this clearer on lines 215-219: ***The vertical profile of the cirrus is investigated using the DAR-DAR dataset, a combination of the CloudSat radar and CALIPSO lidar. The overpass locations of DARDAR are matched within the hour to the TSC at that location, at 1 by 1 degree resolution. A 1°by 1°gridbox can contain many DARDAR retrievals, each of these DARDAR retrievals is assigned the same TSC value - that of the 1°by 1°gridbox. These DARDAR retrievals then all contribute equally to the analysis in the relevant TSC bin.***
* * *
***Line 193****: What is the resolution of the CERES data? The native footprint of ceres is 20 km. Is the CERES data averaged? Also, CERES passes over the tropics 14 times per day on MODIS. Does it provide global coverage in a day? ISCCP H has 3 hourly resolution. How are these data products merged?*

**Reply**: The CERES SYN-1deg dataset is at a 1 by 1° resolution. CERES data provides global coverage at 1 hourly temporal resolution by combining MODIS satellite data with other geostationary satellites. The data products are merged by sampling the CERES data every 3 hours to coincide with the ISCCP data time. The wording of the Data section (lines 94-101) is improved to make this clearer: ***Observational data from the 3-hourly International Satellite Cloud Climatology Project (ISCCP) H dataset at 1x1° is used to define locations of deep convection (Rossow et al., 2017). ECMWF ERA5 reanalysis wind fields are used in the trajectory analysis (Hersbach et al., 2018) and to characterise cloud properties. To examine the evolution of the radiative properties, the CERES SYN1deg L3 LW and SW TOA fluxes are used (NASA/LARC/SD/ASDC, 2017). The CERES SYN1deg product combines MODIS and geostationary satellite data to provide global coverage at a 1x1° resolution and 1 hourly temporal resolution. The vertical evolution of the cloud is investigated by utilising the DARDAR dataset, which is an ice cloud re-***

*trieval product that combines measurements from the CloudSat radar and CALIPSO lidar (Delanoë and Hogan, 2008; Sourdeval et al., 2018). The period of study in this paper is 2008-2010 inclusive.*
* * *
**Line 224**: *If the mean TSC is 180 hours, it would seem that all air has been in contact with convection by day 5. At some point it seems impossible to associate the clouds with convection.*

**Reply**: We are not entirely sure what the reviewer means by this. The reviewer makes a valid point that it becomes more difficult to associate the clouds with convection at longer times since convection, and we have been conscious of not attributing changes in cloud properties at longer TSC to the properties of the initial convective event itself. To this end, we have performed new analysis that explicitly differentiates between gridboxes that would contain detrained cirrus verses those containing cirrus that form in situ along the trajectories. **This is included in section 2.4 and Figure 3b). This analysis shows how the distribution of detrained v in situ cirrus changes as a function of TSC.**
* * *
**Line 227**: *The authors claim that the TSC is relevant to 360 hours. I don't seem how this can be surmised. the air moves more and more into regions of subsidence with time. large-scale subsidence may be as much responsible for the change as some sort of post convection decay of anvil cirrus. Can this possibility be discounted? I think the tendency for the cirrus move out of the tropics with time into the large-scale subsidence as illustrated in Figure 5 is one of the more interesting results presented in the paper.*

**Reply**: The reviewer is correct that large scale subsidence may be responsible for the change rather than convective decay, particularly at long timescales. We were trying to to show how TSC can be used as a metric to characterise changes in cloud properties along a trajectory, not that the properties are changing due to the initial convection itself. The word "relevant" has been removed and L243 has been reworded to emphasise that TSC is merely a useful metric for changes in cloud properties. In addition, section 3.5 has been added to the analysis that includes the zonally averaged DARDAR cloud amount, isolating the dependence of cloud amount on TSC from latitude. Future work will assess the impact of the initial convection on the evolution of high clouds as a function of TSC.

We note that latitudinal variations are not purely responsible for the observed changes in cloud properties. Figure R5 shows the zonally averaged vertical cloud amount for increasing TSC bins. It is clear that at very low TSC values, the vertical cloud fraction is almost only a function of TSC, with very little variation in cloudiness profiles as a function of latitude (Fig. R1a). The cloud amount is consistent across all latitudes for a given TSC at early TSC values, showing that TSC is the controlling factor in the vertical evolution of clouds along TSC trajectories, and not just latitude. Latitude becomes important at longer TSC values, where the TSC becomes less central a controlling factor for the vertical structure of high clouds. This plot has been included in the original paper.
* * *
[Figure]

Figure R5: Zonally averaged DARDAR cloud fraction for 12 hour TSC bins.

**Line 265**: *The authors need to explain how the DARDAR product is able to deduce Ni when the anvils are optically thick for up to 1-day TSC? This implies that only the radar is able to provide credible information to the retrieval. Even when optically thin, the authors need to comment on the validity and uncertainty of the Ni retrieval given that that the lidar is a cross sectional area constraint and the radar a mass-squared constraint. With the shape of the ice crystals unknown, converting the remote sensing into Ni is fraught with uncertainty. Might it be possible that any Ni signal the authors infer are due to changes in ice crystal habit over time? An evolution in habit as the anvils age would result in a time-dependent ratio of mass and area that is not accounted for in DARDAR.*

**Reply**: Producing a new validation of the DARDAR product is out of scope for this work, but we note that for clouds colder than about -40C, the DARADR Ni retrieval compares very favourably to in-situ aircraft measurements (Sourdeval et al., 2018) While the accuracy is reduced in warmer clouds, these are not the primary focus of the study.

However, as the reviewer notes, there are still considerable uncertainties surrounding the retrieval, particularly with particle shape. Although this is approximately a function of temperature, variations in shape over time could produce an apparent change in Ni. While this is planned to be investigated in future work, we have added a note to explain these caveats in the discussion at lines 399-403: *While there are significant changes in DARDAR ice properties as a function of TSC (Fig. 8), there are still considerable uncertainties surrounding the retrieval, particularly at warmer temperatures. The DARDAR-Nice Ni retrieval compares favourably to in-situ measurements at the cirrus temperatures that are the focus of this work (Sourdeval et al., 2018). However, the potential tempo-*

*ral variation of factors such as particle shape might introduce TSC-dependent biases in the Ni retrieval, producing an apparent change in Ni. This is will be investigated in future work through a comparison with aircraft data.*

**Bibliography**

Delanoë, J. and Hogan, R. J.: A variational scheme for retrieving ice cloud properties from combined radar, lidar, and infrared radiometer, Journal of Geophysical Research: Atmospheres, 113, https://doi.org/10.1029/2007JD009000, 2008.

Hersbach, H., Bell, B., Berrisford, P., Biavati, G., Horányi, A., Muñoz Sabater, J., Nicolas, J., Peubey, C., Radu, R., Rozum, I., Schepers, D., Simmons, A., Soci, C., Dee, D., and Thépaut, J.-N.: ERA5 hourly data on pressure levels from 1959 to present., https://doi.org/10.24381/cds.bd0915c6, 2018.

Klein, S. A. and Jakob, C.: Validation and Sensitivies of Frontal Clouds Simulated by the ECMWF Model, Monthly Weather Review, 127, 2514 – 2531, https://doi.org/https://doi.org/10.1175/1520-0493(1999)127⟨2514:VASOFC⟩2.0.CO;2, 1999.

Luo, Z. and Rossow, W. B.: Characterizing Tropical Cirrus Life Cycle, Evolution, and Interaction with Upper-Tropospheric Water Vapor Using Lagrangian Trajectory Analysis of Satellite Observations, Journal of Climate, 17, 4541 – 4563, https://doi.org/10.1175/3222.1, 2004.

NASA/LARC/SD/ASDC: CERES and GEO-Enhanced TOA, Within-Atmosphere and Surface Fluxes, Clouds and Aerosols 1-Hourly Terra-Aqua Edition4A, URL `10.5067/TERRA+AQUA/CERES/SYN1DEG-1HOUR_L3.004A`, 2017.

Rossow, W., Golea, V., Walker, A., Knapp, K., Young, A., Hankins, B., and Inamdar, A.: International Satellite Cloud Climatology Project (ISCCP) Climate Data Record, H-Series, https://doi.org/10.7289/V5QZ281S, 2017.

Sourdeval, O., Gryspeerdt, E., Krämer, M., Goren, T., Delanoë, J., Afchine, A., Hemmer, F., and Quaas, J.: Ice crystal number concentration estimates from lidar–radar satellite remote sensing – Part 1: Method and evaluation, Atmospheric Chemistry and Physics, 18, 14 327–14 350, https://doi.org/10.5194/acp-18-14327-2018, 2018.

Tselioudis, G., Rossow, W. B., Jakob, C., Remillard, J., Tropf, D., and Zhang, Y.: Evaluation of Clouds, Radiation, and Precipitation in CMIP6

Models Using Global Weather States Derived from ISCCP-H Cloud Property Data, Journal of Climate, 34, 7311 – 7324, https://doi.org/10.1175/JCLI-D-21-0076.1, 2021.

---

## Author Response (AR3)

**Reviewer 1**

*: The study by Horner and Gryspeerdt examining the evolution of tropical cirrus as function of time since interaction with deep convection introduces a new method for determining the time displacement since air parcels in the upper troposphere have been in contact with deep convection. While there do seem to be some uncertainties with the method associated with the injection height and the wind used for transport, their general approach seems reasonable. The authors find that cirrus continue to have an important radiative heating in the upper troposphere well beyond the time of injection by deep convection with some signal still evident beyond 120 hours. The authors also examine the properties of these clouds using the new ISCCP dataset, Cloudsat and CALIPSO-derived cloud properties, and CERES cloud radiative effect data sets. In general, I find the study to be very well done with the conclusions supported by the evidence presented in the paper.*

**Reply**: We thank the reviewer for their helpful comments on our paper. We address their concerns below. Line numbers refer to submitted manuscript, not the diffed version.

**General comments:**

*: My primary concern with the study is the interpretation of the ISCCP data. The authors treat the ISCCP data as a literal rendering of cloud type much as one would interpret active remote sensing data to place clouds accurately in the vertical column. ISCCP retrievals have been interpreted, incorrectly, in this way since the product was first introduced by Rossow and Schiffer in 1989. However, I will quote the conclusion of a paper by Chen and Del Genio in 2009: "ISCCP CTPTAU histograms are neither what they were intended to be (a distribution of highest cloud-top heights) nor what they are sometimes mistaken to be (an actual vertical distribution of clouds),but are instead a hybrid of both." This is especially true for cirrus that very often exist above broken low-level clouds and it is one of the reasons why Klein and Jakob (1999) developed the ISCCP simulator software so that models output could be compared with ISCCP. In Mace and Wrenn (2013), we examined the vertical distribution of cloud layers observed by active remote sensors within the ISCCP CTP-tau histograms. We found that a significant fraction of mid-level clouds (up to half) in the tropics are high-level clouds incorrectly diagnosed to have clout top pressures in the middle instead of upper troposphere. While it is not my opinion that this issue negates the central findings of this study, the authors need to consider their interpretation of Figure 6. It is my opinion that the increase in mid-level clouds seen in successive periods in Figure 6 could be caused by continual misdiagnosis of cirrus as mid-level clouds. This would occur when optically thin high clouds move over the greater low level cloud fractions in the trade cumulus regions on the poleward reaches of the tropics. The CRE results in figure 10 also need to be reconsidered as underestimating the effect of thin cirrus. At the very least,*

*the authors need to comment on this issue and acknowledge the potential for the ISCCP data to not provide a literal rendering of cloud top pressure or optical depth.*

**Reply**: We are very grateful to the reviewer for these helpful comments in diagnosing the presence of the mid-level clouds. We are aware of these particular issues with the ISCCP data, and have included a paragraph in the discussion section that acknowledges these issues. See lines 415-425: ***Furthermore, it is important to note that the ISCCP dataset can provide a biased representation of cloud top pressure and cloud optical depth. As stated in Chen and Del Genio (2009), the ISCCP CTP-TAU histograms are not an actual vertical distribution of clouds. This means that when considering Fig. 6., the apparent cloud layer that appears at 370 hPa should not be taken to be a true appearance of the clouds at this height. More likely, it is the existence of thin high clouds sitting over low-cloud that causes ISCCP to incorrectly diagnose them as lower level clouds (Mace and Wrenn, 2013). This occurs when the optically thick deep convection dissipates, and the thin cirrus moves over a region of continuous low cloud in the trade cumulus regions. The issue with the ISCCP mid level clouds is not concerning, and should not impact our subsequent results, in particular the CRE calculations in Fig. 10. To isolate our high cloud regions for the CRE calculations we select bins with very little low clouds, making it unlikely that this mid-level cloud effect drives the results in Fig. 10.***
* * *
*: I also wonder why the authors chose to present the ice crystal number concentration (Ni) in the DARDAR results in Figure 8? Ni is the least reliable parameter of this retrieval and probably represents the least understood parameter and the least interesting in my opinion. Of much more interest would be the evolution of the ice water content and effective radius since those parameters are directly related to the radiative effects of the cirrus. Given the general decrease in Ni with time in the cirrus layer, I wonder if the water content is decreasing (as it should given the optical thickness trend) or if the effective radius is changing with time, or both? One would also expect there to be some amount of size sorting with larger effective radii lower in the cirrus layer. Results using the CloudSat 2C-ICE data set should also be generated and compared with the DARDAR results since both are retrievals with assumptions and likely have considerable uncertainties. The two algorithms, while using the same inputs, are independent in the manner that the retrievals are conducted.*

**Reply**: The reviewer makes a useful recommendation to consider analysing other microphysical properties from DARDAR and CloudSat 2c-ice data. In this paper, Ni was used merely to illustrate the use of the DARDAR data for investigating ice microphysical properties. This analysis is planned for future work that will further examine the behaviour of cloud properties as a function of TSC, where we intend to consider the useful suggestions made here.

**Reviewer 2**
* * *
*: I appreciate the authors' additional anlysis to differentiate between detrained cirrus and in-situ cirrus that form after the detrained cirrus dissipated. However, I am concerned that the method used to do this is not correct. The authors stated that whenever the cloud fraction of the detrained cirrus has dropped below 10% for the first time, then any subsequent cirrus are considered to form in situ. The authors stated that this method is similar to that used by Luo and Rossow (2004), but I don't think that this is the case.*

**Reply**: We thank the reviewer for their helpful comments on our paper. It is evident that we should be more clear in stating that whilst we do believe our method to be similar to Luo and Rossow (2004), it is not the same method. As mentioned by the reviewer, Luo and Rossow (2004) define the end of lifetime for the detrained cirrus (i.e. their 'zero cirrus' case), to be 1/5 of the maximum cirrus cloud amount. In our paper, we define the 'zero cirrus' case to be a 10% cloud fraction. In practice, these methods are very similar, because the maximum cloud amount along a trajectory is, due to the definition used for deep convection, very close to 100% cloud fraction. So a 10% cloud fraction is not significantly different from 10% of the maximum cloud fraction. We have rephrased lines 163-164 to make it clear that our method is similar, but not identical to, Luo and Rossow (2004): ***This is a similar method used in Luo and Rossow (2004), who use a threshold of 20% of the maximum cirrus cloud fraction along a trajectory to identify the 'zero detrained cirrus' case, rather than a 10% cloud fraction threshold as used in this work.***

**General comments:**
* * *
*: Figures 8(a) and 8(b) of this manuscript show that the cloud fraction at TSC = 0 is 0.3 on average. Therefore, if we apply the method used by Luo and Rossow (2004), then on average the detrained cirrus would disappear when the cloud fraction is reduced below 0.3/5 = 0.06 = 6%, not 10%. Moreover, the maximum cloud fraction obtained at TSC = 0 may be different for different trajectories, so the threshold to determine the disappearance of the detrained cirrus would change from one trajectory to the next.*

**Reply**: The reviewer is using the cloud amount from DARDAR in Figure 8(a) and 8(b), however in our paper the cloud fraction used to determine whether the threshold has been met is from ISCCP. The maximum cloud fraction from ISCCP is, by design, very close to 100% for TSC=0, as this is a requirement for the identification of deep convection. This data source has been explicitly stated on line 160: ***Once the ISCCP cloud fraction drops below 10% along a given trajectory[...]***
* * *
*: Figure 3(b) does not make sense to me, either. For TSC less than about 10–15 hours, the number of counts of in-situ cirrus is not visible because it is shown*

*beneath the number of counts of detrained cirrus, but let me assume that in this figure these two numbers are equal for TSC less than about 10–15 hours. If this is what's plotted in the figure, I don't understand why there would be so many in-situ cirrus already for small TSC.*

**Reply**: The histogram in 3(b) is additive, i.e. the detrained component and in situ component are stacked on top of each other. We appreciate this isn't immediately clear, so have amended the histogram so that the detrained and cirrus components are overlaid in the histogram. As is hopefully clear from this histogram, there is significantly more detrained cirrus than in situ cirrus at small TSC.
* * *
*: The authors aim to demonstrate that convection has a significant long-lasting impact on the properties of clouds. However, in Fig. 7(b), the clouds at 370 hPa do not appear to be connected to the initial convection. Please see also my comment on the original version of the manuscript about these clouds. It appears to me from Fig. 7(b) that these clouds are neither convectively detrained nor formed in-situ from the moisture perturbation brought about by the convection. The presence of these clouds certainly affects the average cloud properties, for example, cloud radiative effect (CRE). It follows that the CRE shown in Fig. 10 is not purely from clouds associated with convection and cannot be used to demonstrate the impact of convection on CRE. I believe this is a major issue with this study.*

**Reply**: The clouds in this 370hPa band are an artifact of both the ISCCP histogram bin aliasing to those used in Fig. 7b), and a retrieval bias in the ISCCP dataset, rather than indicating any true cloud that exists. Note that this bin aliasing doesn't appear on Fig.7a) as the wider bins are less likely to suffer from aliasing effect. The retrieval bias means the ISCCP histograms underestimate the thin high cloud that sits over low cloud, mischaracterising them instead as mid-level clouds (Mace and Wren, 2013). This retrieval bias maybe also contribute to the increase in the mid level cloud at 370hPa. They become more visible as the optically thick convective clouds dissipate. These results should not affect the CRE in Fig. 10. significantly, as we define the presence of high cloud as a deficit of low cloud. This means we aren't relying on the amount of high cloud (which we believe to be underestimated) to define where we expect high cloud to be in the high cloud CRE. We appreciate the need for clarifcation on this point, therefore we have included an extra paragraph in the discussion section to address the shortcomings of the ISCCP dataset on lines 416-425:

**Furthermore, it is important to note that the ISCCP dataset does not necessarily provide a true representation of cloud top pressure or cloud optical depth. As stated in Chen and Del Genio (2009), the ISCCP CTP-TAU histograms are not an actual vertical distribution of clouds. This means that when considering Fig. 6., the apparent cloud layer that appears at 370 hPa should not be taken to be a true appearance of the clouds at this height. More likely, it is the existence of thin high clouds sitting over low-cloud that causes ISCCP to incor-**

*rectly diagnose them as lower level clouds (Mace and Wrenn, 2013). This occurs when the optically thick deep convection dissipates, and the thin cirrus moves over a region of continuous low cloud in the trade cumulus regions. The DARDAR dataset and Fig. 8. is particularly useful here as it can provide us with a more reliable vertical distribution of the cloud amount. The issue with the ISCCP mid level clouds is not concerning, and should not impact our subsequent results, in particular the CRE calculations in Fig. 10. To isolate our high cloud regions for the CRE calculations we select bins with very little low clouds, making it unlikely that this mid-level cloud effect drives the results in Fig. 10.*
* * *
*: Finally, a specific issue to be fixed is the caption of Fig. 9. In the current version of the manuscript, it is incorrectly identical to the caption of Fig. 10*

**Reply**: We thank the reviewer for raising this specific issue. This has been fixed, with a new caption included for Figure 9.: *Zonally averaged DARDAR vertical cloud amount for a given TSC bin. TSC bins are given in the top right corner.*

---

## Author Response (AR4)

**Editor Comments**

*: Dear Dr. Horner,*

*Thank you for the adjustments made to you manuscript in response to the thoughtful reviewer comments. I have one further recommendation prior to publication following on the comment from Jay Mace about ice crystal concentrations. In the methods section it would be beneficial I believe if a few words of justification were given for why the ice crystal number concentration is reported given that indeed Nice is quite difficult to retrieve and that other quantities such as ice water content might be seen as more reliable. Otherwise, I look forward to publication of your interesting manuscript about the evolution of cirrus outflows. Congratulations.*

*Best regards,*

*Tim Garrett*

**Reply**: We thank the reviewer and editor for all of their help during the publication of this manuscript. We address the final revision from the editor on lines 100-104: ***To illustrate the vertical cloud evolution and microphysical properties the DARDAR dataset, an ice cloud retrieval product that combines measurements from the CloudSat radar and CALIPSO lidar (Delanoë and Hogan, 2008; Sourdeval et al., 2018), is used.*** $N_i$ ***is highlighted from the DARDAR data due to the complimentary information it provides about cloud history compared to the large scale properties from ISCCP (Krämer et al., 2016).***